# Mobility-LLM: Learning Visiting Intentions and Travel Preferences from Human Mobility Data with Large Language Models

**Letian Gong**[1,2]*, **Yan Lin**[3]*, **Xinyue Zhang**[1,2], **Yiwen Lu**[1], **Xuedi Han**[1]
**Yichen Liu**[1,2], **Shengnan Guo**[1,2]†, **Youfang Lin**[1,2], **Huaiyu Wan**[1,2]
[1]School of Computer Science and Technology, Beijing Jiaotong University, China
[2]Beijing Key Laboratory of Traffic Data Analysis and Mining, Beijing, China
[3]Department of Computer Science, Aalborg University, Denmark
{gonglt, zhangxinyue, luyiwen, hanxuedi, liuyichen, guoshn,
yflin, hywan}@bjtu.edu.cn, lyan@cs.aau.dk

## Abstract

Location-based services (LBS) have accumulated extensive human mobility data on diverse behaviors through check-in sequences. These sequences offer valuable insights into users' intentions and preferences. Yet, existing models analyzing check-in sequences fail to consider the semantics contained in these sequences, which closely reflect human visiting intentions and travel preferences, leading to an incomplete comprehension. Drawing inspiration from the exceptional semantic understanding and contextual information processing capabilities of large language models (LLMs) across various domains, we present Mobility-LLM, a novel framework that leverages LLMs to analyze check-in sequences for multiple tasks. Since LLMs cannot directly interpret check-ins, we reprogram these sequences to help LLMs comprehensively understand the semantics of human visiting intentions and travel preferences. Specifically, we introduce a visiting intention memory network (VIMN) to capture the visiting intentions at each record, along with a shared pool of human travel preference prompts (HTPP) to guide the LLM in understanding users' travel preferences. These components enhance the model's ability to extract and leverage semantic information from human mobility data effectively. Extensive experiments on four benchmark datasets and three downstream tasks demonstrate that our approach significantly outperforms existing models, underscoring the effectiveness of Mobility-LLM in advancing our understanding of human mobility data within LBS contexts.

## 1 Introduction

Location-based services (LBS) such as Gowalla, Weeplace, and Foursquare enable users to share and discover location information and nearby services. This results in the collection of extensive human mobility data, often presented in the form of check-in sequences. These sequences record users' visits to different points of interest (POIs) like restaurants and hospitals at various times, reflecting significant semantics about their intentions and preferences. Analyzing these check-in sequences is crucial as it offers valuable information on human mobility data, which can positively impact individuals, businesses, and urban management.

---

*Both authors contributed equally to this research.
†Corresponding author.

38th Conference on Neural Information Processing Systems (NeurIPS 2024).

The key to effectively mining check-in sequences lies in understanding their rich semantics. Existing methods primarily focus on specific tasks, such as location prediction [10, 59, 52, 26], time prediction [40, 45], and trajectory user linking [34, 13, 62], rather than delving into the semantics of human behaviors. This narrow focus often results in limited optimization goals and a shallow understanding of the semantics contained in check-in sequences. Recently, large language models (LLMs) have demonstrated impressive capabilities in semantic understanding and contextual information processing, demonstrating successful adaptability across different domains. LLMs trained on extensive corpora surpass task-specific models in their potential to understand semantic information. Inspired by this, we aim to utilize pre-trained LLMs as powerful check-in sequence learners.

Nevertheless, LLMs encounter a significant obstacle in their inability to directly interpret check-in sequences. As typical sequential data, check-in sequences contain a wealth of semantic information that reflects various near-term regularities and inherent characteristics. The future intention of an individual is prone to be dictated by near-term regularities that are close to recent visits, termed *visiting intentions*. Furthermore, an individual's inherent characteristics tend to persist over time and determine their *travel preferences*, which is necessary to analyze them across multiple domains for a comprehensive understanding. Hence, our main challenge is to enable LLMs to effectively extract semantics from check-in sequences and comprehensively understand human visiting intentions and travel preferences.

To address this challenge, we present a novel unified framework called **Mobility-LLM** for various check-in sequence analysis tasks. It leverages pre-trained LLMs for general check-in sequence analysis. Our contributions can be summarized as follows:

- We propose a unified framework called **Mobility-LLM** that uses a pre-trained LLM to achieve a SOTA or comparable performance across various check-in analysis tasks including location prediction, trajectory user link, and time prediction. We extract the semantics of check-in sequences to enable LLMs to gain a comprehensive understanding of human visiting intentions and travel preferences.
- A visiting intention memory network (VIMN) is proposed for capturing users' visiting intentions of users at each check-in record by prioritizing relevant check-in records.
- A shared pool of human travel preference prompts (HTPP) in different domains is introduced, which enables a comprehensive understanding of human travel preferences and matches appropriate prompts from multiple domains.
- Our model's exceptional performance is validated through extensive experiments on four benchmark datasets involving three tasks. Our robust outcomes in cross-domain pre-training exhibit an average enhancement of 17.8% and an average of 23.6% to 38.3% on the few-shot scenario.

## 2 Related Works

In this section, we provide short reviews of literature in the areas of mobility data mining and cross-domain applications of LLMs. We postpone the detailed discussion of works to the Appendix A, due to limited space.

**Mobility Data Mining** has emerged as a promising research area due to the proliferation of location-based services. This has led to the development of three significant tasks that enhance service quality: next location prediction (LP), next time prediction (TP), and trajectory user link (TUL). The LP task aims to anticipate a user's future location based on their historical movement. Several notable models have emerged as leading approaches in LP [10, 59, 51, 48, 5, 23, 58]. The TUL task focuses on establishing connections between different trajectories and users, facilitating the analysis of user movement patterns and uncovering valuable insights into their behavior [13, 7, 60, 16]. The TP task focuses on estimating the time at which a user is likely to visit their next location. Various models have been developed to model the intensity function representing the rate or density of event occurrences, effectively making accurate time predictions [57, 17, 53].

**Cross-domain Application of LLMs** has gained attention in recent studies, which adopt large language models to address the challenge of limited training data. In the field of time series analysis, LLM4TS [4] is a pioneering method that aligns pre-trained large language models with temporal characteristics, introducing a two-level aggregation method to effectively incorporate multi-scale temporal data into pre-trained LLMs. One-Fits-All [63] is a unified framework that leverages a frozen

pre-trained language model to attain state-of-the-art or comparable performance across various major types of time series analysis tasks. AutoTimes [30] facilitates the tokenization of time series into the embedding space of LLMs and intelligently utilizes the inherent token transitions to effectively predict time series in an autoregressive manner. In the field of computer vision, LM4VE [35] incorporates a frozen transformer block from an LLM as a general-purpose visual encoder layer. To tackle graph-related tasks, TAPE [19] utilizes semantic knowledge generated by LLMs to enhance the quality of initial node embeddings in GNNs. MoleculeSTM [27] aligns GNNs and LLMs within a shared vector space, integrating textual knowledge into graphs to enhance reasoning capabilities.

## 3 Preliminaries

**POI Visiting Record** In the check-in datasets, a user's visit to a certain place is represented by a POI visiting record $R = (L_p, t)$ generated by the user $u$. $L_p$ indicates the visited POI at time $t$. $L_p$ is represented by $(L_{id}, L_{lon}, L_{lat}, L_{category})$, comprising $L_{id}$ as a POI index, and accurate longitude $L_{lon}$ and latitude $L_{lat}$. $L_{category}$ denotes the category of the visited POI (e.g., hospital, restaurant).

**Check-in Sequence** A user's movement during a specific period can be represented by sequential POI visiting records, which we refer to as a check-in sequence. We denote a check-in sequence as $\mathcal{C} = < R_1, R_2, \cdots, R_n >$, where the POI visit records are ordered by their visited time, and $n$ is the length of the sequence.

**Problem Statement** Given a check-in sequence $\mathcal{C}$, our objective is to encode this sequence into a meaningful representation. This representation can be used for various tasks. In this paper, we choose three typical check-in prediction tasks: 1) Identifying the user $\hat{u}$ who generated the check-in sequence (TUL task). 2) Predicting the next location $\hat{s}_{n+1}$ the user will arrive at (LP task). 3) Predicting the arrival time $\hat{t}_{n+1}$ at this location (TP task).

## 4 Methodology

We introduce a novel unified framework, **Mobility-LLM**, designed to address a variety of check-in sequence tasks. The overall structure of this framework is illustrated in Fig. 1. 1) Initially, to embed POIs by incorporating category semantics, we introduce the POI Point-wise Embedding Layer (PPEL) to generate the embedding of POIs in the current check-in record (referred to as PPE). 2) Subsequently, we feed the PPEs and timestamps of a check-in sequence into the Visiting Intention Memory Network (VIMN) to capture the visiting intentions of users at each check-in record. 3) A Human Travel Preference Prompt (HTPP) pool is introduced to extract users' preferences from check-in sequences, which act as cues to assist the LLM in comprehending users' travel preferences more effectively. 4) Finally, we use the different parts outputs of LLM (corresponding to VIMN, HTPP), each with its own projection head, to forecast the user's next location, the estimated arrival time, and the user who generates the check-in sequence.

**POI Point-wise Embedding Layer (PPEL)** is designed to generate the semantic information embedding for each POI in a check-in record. This is especially important since POI categories contain a wealth of semantic information. It has been observed that the category descriptions of POIs in the original check-in datasets are often vague or unidentified. When we refer to "vague," we are indicating that the descriptions are too broad or contain abbreviations, making it challenging to determine the specific POI categories accurately. To address this issue, we have developed a category word pool (see Appendix I for the whole categories list) that allows each POI to match with the most appropriate categories automatically. We reindex the POI IDs and word IDs in each dataset, making it convenient for subsequent encoding of these IDs. As shown in Fig. 1a, we use the learnable embedding $E_{L_{id}}$ of each POI ID as the query, the learnable embedding $E_{C_{id}}$ of the category word ID as the key, and the corresponding category word token $E_{C_{\text{token}}}$ from the LLM tokenizer as the value. A Point-wise attention mechanism is used to calculate the $i$-th PPE $\boldsymbol{s}_i$ of POIs:

$$\boldsymbol{s}_i = \text{Softmax}(\frac{\text{Que}(E_{L_{id}})\text{Key}(E_{C_{id}})^T}{\sqrt{d}})\text{Val}(E_{C_{\text{token}}}) + \text{GeoHash}(L_{lon}, L_{lat}), \quad (1)$$

where $\text{Que}(\cdot)$, $\text{Key}(\cdot)$, and $\text{Val}(\cdot)$ denote linear projections, $d$ is the dimension of $E_{C_{\text{token}}}$, $\text{GeoHash}(\cdot, \cdot)$ encodes geographic coordinates (latitude and longitude) into an embedding vec-

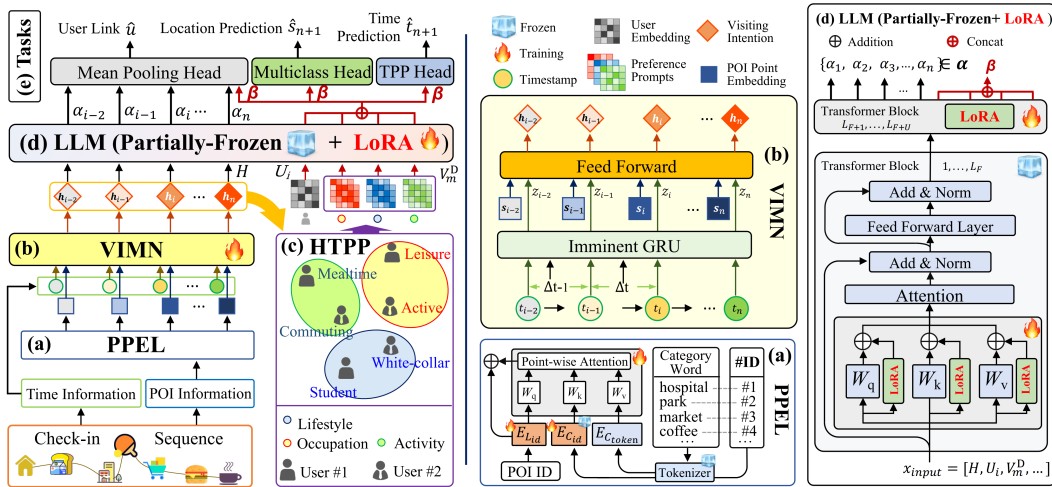

Figure 1: The overall of our Mobility-LLM framework. a) POI Point-wise Embedding Layer (**PPEL**). b) Visiting Intention Memory Network (**VIMN**). c) Human Travel Preference Prompt (**HTPP**). d) $\alpha$ denotes the output of the LLM corresponding to VIMN (i.e. first $n$ output of the LLM), while the remaining outputs are denoted as $\beta$.

tor (see Appendix H for details). We omit the MLP Layer and Layer Normalizations in Transformer layers to simplify the representation in Eq. 1.

**Visiting Intention Memory Network (VIMN)** is proposed to capture the visiting intentions of users at each check-in record by prioritizing relevant check-in records. As shown in Fig. 1b, we feed the timestamp of each check-in record and the time interval between the adjacent records into the Imminent GRU layer. We adopt a dual encoding approach for time representation: 1) Periodic encoding for timestamps $t$ as $T(t) = [\cos(\omega_1 t), \sin(\omega_1 t), \ldots, \cos(\omega_k t), \sin(\omega_k t)]$, where $\{\omega_k\}$ are frequencies determined to capture periodicity across various temporal scales. 2) Logarithmic encoding for time intervals $\Delta t$ represented as $\Delta T = \log(1 + \Delta t)$, which adjusts the GRU's forget gate based on the time interval $\Delta t$. This adjusted factor, denoted as $\Delta T$, affects the forget gate of the GRU unit through $G_{\text{forget}}(\Delta T) = \sigma(W_f \Delta T + b_f)$. The other components like $G_{\text{update}}$ remain unchanged from the original GRU configuration. The Imminent GRU layer can be depicted as:

$$z_i = \sigma(W_{in} T(t_i) + G_{\text{update}}(\mathcal{H}_{i-1}) \times G_{\text{forget}}(\Delta T)), \tag{2}$$

where $z_i$ represents the output of the Imminent GRU at time step $t_i$, and $\mathcal{H}_{i-1}$ denotes the hidden state at time step $i-1$. This setup allows for filtering out less relevant, temporally distant data. Subsequently, the outputs $[z_{i-r+1}, \cdots, z_{i-1}, z_i]$ from the most recent $r$ cycles of the Imminent GRU, along with the latest $r$ PPEs, are forwarded to the feed-forward layer [44] (further elaborated in Appendix G) to refine the representation of the user's visiting intentions $h_i$.

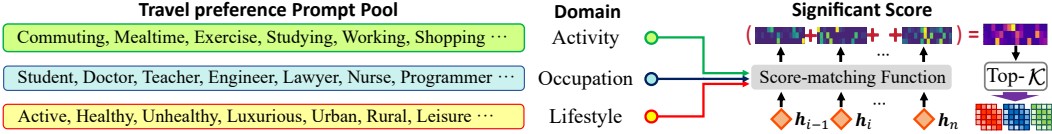

Figure 2: The architecture of HTPP.

**Human Travel Preference Prompt (HTPP).** This paragraph introduces a method for extracting users' travel preferences to help the LLM enable a comprehensive understanding of human travel preferences and match appropriate prompts from multiple domains. Prompting strategies have demonstrated encouraging outcomes in various applications that aid model predictions [3]. Previous works primarily focus on utilizing a fixed prompt to boost the pre-trained models' performance through fine-tuning [30] or learnable prompts lacking reality semantic meaning [41, 3]. User behavior is diverse and cannot be accurately summarized as a single fixed prompt to describe the user's travel

preferences. Adopting a prompt-based approach with meaningless learnable prompt vectors can only handle sequences with simple semantics, such as time series [41, 3]. However, it lacks the capability to adequately extract human travel preferences. To address this limitation, we introduce a shared pool of travel preference prompt words across $D = 3$ domains (e.g., occupation, activity type, lifestyle) as illustrated in Fig. 2. Each domain includes a selection of $m = 16$ prompt words. For every prompt word, we create a key-value pair where the value represents the word token obtained from the LLM tokenizer, and the key is derived from the word token through a trainable linear transformation. Specifically, these key-value pairs are defined as follows:

$$P = \{(\boldsymbol{k}_1^D, V_1^D), (\boldsymbol{k}_2^D, V_2^D), \cdots, (\boldsymbol{k}_m^D, V_m^D)\}, D \in \{1, 2, 3\}, \tag{3}$$

where $\boldsymbol{k}_m^D, V_m^D \in \mathbb{R}^{L_E}$ is a key-value pair, and we maintained it at the same embedding size $L_E$ as visiting intentions $\boldsymbol{h}_i$. We then employ a scoring function $\phi(\boldsymbol{h}_i, \boldsymbol{k}_m^D) = \boldsymbol{h}_i \cdot \boldsymbol{k}_m^D / \|\boldsymbol{h}_i\| \|\boldsymbol{k}_m^D\|$ to determine the significance score of each $\boldsymbol{h}_i$, where $\phi : \mathbb{R}^{L_E} \times \mathbb{R}^{L_E} \to \mathbb{R}$. The significant score of each $\boldsymbol{h}_i$ indicates the relevance of the current visiting intention to each word in the travel preference prompts pool. Subsequently, we aggregate the significant scores of each $\boldsymbol{h}_i$ to identify the top-$\mathcal{K}$ most significant pairs in a domain. By defining $\{m_j\}_{j=1}^{\mathcal{K}}$ as a subset of indices for the selected top-$\mathcal{K}$ prompt words in each domain, we obtain the output $[V_{m_1}^D; \cdots; V_{m_{\mathcal{K}}}^D]$ of HTPP as prompts (illustrated in Fig. 1c) to more accurately recognize and comprehensively understand user's travel preferences at the level of individual check-in sequences.

**Training Task**  In Fig. 1d, the input to the LLMs consists of the output of VIMN $H = [\boldsymbol{h}_1, \boldsymbol{h}_2, \cdots, \boldsymbol{h}_n]$, HTPP, and the user embedding $U_i$, denoted as $\boldsymbol{x}_{input}$:

$$\boldsymbol{x}_{input} = \left[H, U_i, V_{m_1}^D; \cdots; V_{m_{\mathcal{K}}}^D\right], D \in \{1, 2, 3\} \tag{4}$$

We concatenate all the tokens along the check-in length dimension. As shown in Fig. 1e, $\boldsymbol{\alpha}$ denotes the output of the LLM corresponding to VIMN (i.e., the first $n$ output of the LLM), while the remaining outputs are denoted as $\boldsymbol{\beta}$, where $n$ is the length of this check-in sequence. Various projection heads are used to predict the next location $\hat{s}_{n+1}$ the user will arrive at and its arrival time $\hat{t}_{n+1}$ with the $\boldsymbol{\beta}$. We use the $\boldsymbol{\beta}$ and $\boldsymbol{\alpha}$ with a mean pooling projection head to predict the user $\hat{u}$ who generates this check-in sequence.

# 5  Experiments

To evaluate the performance of our proposed model, we carry out extensive experiments on four real-world check-in sequence datasets, targeting three different types of downstream tasks: Next Location Prediction (LP), Trajectory User Link (TUL), and Time Prediction (TP). We use TinyLlama-1B [56] as the default backbone unless stated otherwise. The code of Mobility-LLM is released at *https://github.com/LetianGong/Mobility-LLM*.

**Baselines:** For the LP task, we cover five state-of-the-art LP models to demonstrate the superiority of our model: DeepMove [10], LightMove [22], PLSPL [47], HMT-LSTM [25], LSTPM [59]. For the TUL task, we select four end-to-end models for comparison: TULER [13], TULVAE [62], MoveSim [11], S2TUL [7]. For time prediction methods, we select four SOTA models for comparison: IFLTPP [40], THP [64], NSTPP [17], DSTPP [53]. NSTPP and DSTPP can also be applied to the LP task. For sequence representation methods ReMVC [55], VaSCL [54], SML [61], CACSR [15], we apply them to learn the representation of the check-in sequence and serve different downstream tasks. More details are in Appendix D.

**Datasets:** In our experiments, we use four real-world datasets derived from Gowalla [3], WeePlace [28, 31], Brightkite [4], and FourSquare [49, 50]. To ensure data consistency, we set a maximum historical time limit of 120 days and filter out users with fewer than 10 records and places visited fewer than 10 times. Appendix C provides statistical information for each processed dataset. We shuffle all the samples and then split the datasets into training, validation, and test sets in a 6:2:2 ratio based on the number of samples.

---

[3]https://snap.stanford.edu/data/loc-Gowalla.html
[4]https://snap.stanford.edu/data/loc-brightkite.html

**Implement Details:** During the training phase, we adopt a partially frozen strategy to fine-tune the pre-trained LLM. We apply different parameter freezing strategies to the $1 - L_F$ layers and the $L_F - L_{F+U}$ layers. In order to enhance the model's performance on trajectory data, we employ the Low-Rank Adaptation (LoRA) algorithm [21] to incorporate additional parameters into the LLMs. Details of the partially frozen strategy and other settings can be found in Appendix B. We run each set of experiments 5 times and reported their mean values

## 5.1 Next Location Prediction

Table 1: Next location prediction (LP) performance results. A higher value indicates better performance. **Red**: the best, Blue: the second best. The units of all metics are expressed as e-2.

| Datasets | Gowalla | | | | WeePlace | | | | Brightkite | | | | FourSquare | | | |
|---|---|---|---|---|---|---|---|---|---|---|---|---|---|---|---|---|
| Metric (e-2) Method | Acc@1 | Acc@5 | Acc@20 | MRR | Acc@1 | Acc@5 | Acc@20 | MRR | Acc@1 | Acc@5 | Acc@20 | MRR | Acc@1 | Acc@5 | Acc@20 | MRR |
| DeepMove | 10.51 | 23.21 | 33.9 | 16.65 | 19.23 | 37.79 | 52.61 | 27.03 | 49.82 | 66.25 | 71.19 | 56.94 | 16.3 | 35.06 | 48.39 | 25.02 |
| LightMove | 9.88 | 20.93 | 29.95 | 15.01 | 18.33 | 36.37 | 52.75 | 27.03 | 49.01 | 63.11 | 68.94 | 55.46 | 13.27 | 29.33 | 41.45 | 20.71 |
| PLSPL | 11.26 | 24.12 | 33.82 | 17.44 | 18.77 | 37.31 | 53.31 | 27.86 | 51.42 | 65.34 | 71.46 | 57.59 | 13.73 | 30.65 | 43.18 | 21.42 |
| HMT-LSTM | 10.73 | 22.41 | 32.77 | 16.47 | 17.29 | 34.23 | 49.82 | 25.69 | 49.22 | 63.36 | 67.96 | 55.41 | 13.67 | 29.9 | 42.6 | 21.17 |
| LSTPM | 9.83 | 20.88 | 30.25 | 15.12 | 15.6 | 31.15 | 45.98 | 23.34 | 42.58 | 54.65 | 60.21 | 48.16 | 15.46 | 34.17 | 48.49 | 24.27 |
| VaSCL | 11.47 | 22.17 | 32.92 | 16.56 | 18.11 | 37.54 | 53.04 | 28.46 | 49.95 | 66.21 | 71.17 | 57.23 | 14.99 | 32.84 | 47.06 | 23.39 |
| SimCSE | 7.15 | 16.12 | 23.8 | 11.53 | 14.12 | 31.06 | 46.89 | 22.31 | 46.65 | 64.48 | 70.43 | 54.69 | 14.5 | 33.13 | 47.89 | 23.29 |
| NSTPP | 10.81 | 23.23 | 32.94 | 16.87 | 16.58 | 32.37 | 47.82 | 24.2 | 45.83 | 58.44 | 64.56 | 52.16 | 14.89 | 33.18 | 47.03 | 23.36 |
| DSTPP | 10.85 | 23.11 | 33.19 | 16.74 | 18.85 | 37.68 | 53.44 | 27.52 | 48.71 | 62.82 | 67.71 | 55.26 | 13.3 | 29.11 | 41.53 | 20.66 |
| ReMVC | 11.03 | 22.94 | 33.38 | 16.65 | 18.07 | 35.92 | 51.93 | 26.66 | 49.57 | 63.58 | 69.28 | 56.31 | 16.92 | 36.05 | 49.39 | 26.02 |
| SML | 9.92 | 20.91 | 30.36 | 15.25 | 17.42 | 35.23 | 51.07 | 25.96 | 46.26 | 58.93 | 65.35 | 51.97 | 14.72 | 32.54 | 46.87 | 23.27 |
| CACSR | 10.94 | 18.22 | 26.56 | 12.83 | 19.66 | 36.46 | 51.25 | 28.15 | 44.56 | 62.01 | 65.91 | 51.91 | 14.73 | 31.54 | 46.47 | 22.78 |
| **Mobility-LLM** | **11.87** | **25.14** | **36.36** | **18.29** | **20.47** | **39.22** | **56.69** | **29.21** | **53.18** | **68.31** | **74.11** | **59.89** | **17.29** | **37.17** | **53.16** | **26.47** |

Table 2: Trajectory user link (TUL) performance results. A higher value indicates better performance. **Red**: the best, Blue: the second best. The units of all metics are expressed as e-2.

| Datasets | Gowalla | | | | WeePlace | | | | Brightkite | | | | FourSquare | | | |
|---|---|---|---|---|---|---|---|---|---|---|---|---|---|---|---|---|
| Metric (e-2) Method | Acc@1 | Acc@5 | Acc@20 | MRR | Acc@1 | Acc@5 | Acc@20 | MRR | Acc@1 | Acc@5 | Acc@20 | MRR | Acc@1 | Acc@5 | Acc@20 | MRR |
| TULER | 55.85 | 64.55 | 72.19 | 60.09 | 63.69 | 79.24 | 86.4 | 72.37 | 58.37 | 73.48 | 83.36 | 65.26 | 44.43 | 58.41 | 69.14 | 51.04 |
| TULVAE | 41.33 | 43 | 44.6 | 42.22 | 32.72 | 39.02 | 44.87 | 36.22 | 37.76 | 47.42 | 54.99 | 42.68 | 21.7 | 26.52 | 31.21 | 24.44 |
| MoveSim | 46.5 | 59.42 | 68.21 | 52.66 | 57.07 | 70.49 | 79.13 | 63.42 | 60.19 | 70.2 | 78.52 | 65.04 | 37.48 | 50.65 | 61.57 | 44.02 |
| S2TUL | 59.33 | 67.78 | 67.07 | 61.2 | 52.82 | 55.09 | 57.43 | 51.77 | 39.54 | 44.46 | 44.73 | 48.7 | 37.97 | 45.08 | 43.42 | 44.85 |
| VaSCL | 59.9 | 68.34 | 74.99 | 63.99 | 74.31 | 83.25 | 88.64 | 78.42 | 63.88 | 67.67 | 68.67 | 63.86 | 51.44 | 61.22 | 69.72 | 56.38 |
| SimCSE | 26.78 | 43.63 | 57.76 | 35.02 | 55.58 | 71.74 | 82.56 | 63.34 | 59.96 | 71.11 | 79.9 | 65.39 | 40.1 | 53.15 | 65.54 | 46.78 |
| ReMVC | 68.75 | 74.4 | 73.19 | 70.02 | 65.78 | 73.09 | 71.64 | 66.15 | 73.85 | 82.55 | 87.93 | 77.93 | 58.18 | 66.84 | 72.67 | 65.14 |
| SML | 57.42 | 62.44 | 63.07 | 61.97 | 59.44 | 69.93 | 69.52 | 62.43 | 63.91 | 72.28 | 70.52 | 66.38 | 55.69 | 58.42 | 63.7 | 57.7 |
| CACSR | 52.62 | 63.5 | 71.29 | 57.84 | 70.01 | 81.3 | 86.98 | 75.24 | 58.6 | 72.54 | 79.72 | 65.18 | 51.89 | 64.59 | 72.91 | 57.98 |
| **Mobility-LLM** | **80.43** | **86.29** | **88.56** | **83.18** | **79.03** | **88.04** | **91.48** | **83.21** | **83.06** | **88.52** | **90.35** | **85.73** | **72.08** | **79.67** | **84.32** | **75.71** |

**Setups:** We consider the LP task as a multi-classification problem. Given a check-in sequence $\mathcal{C}^{U_i}$ from a specific user $U_i$, we feed it to our framework $G_{LLM}$ to obtain the check-in sequence representation $G_{LLM}(\mathcal{C}^{U_i})$. As shown in Fig. 1e, a multi-class projection head $f_{\boldsymbol{\theta}}^{multi}(\boldsymbol{\beta}) = \text{softmax}(W\boldsymbol{\beta} + b))$ is used to predict the next location $\hat{s}_{n+1}$ with the corresponding output $\boldsymbol{\beta}$ of VIMN. We maximize the conditional log-likelihood for a given $N$ observations as follows: $\mathcal{L}_{MLE}(\boldsymbol{\theta}) = \sum_{i=1}^{N} \log f_{\boldsymbol{\theta}}^{multi}(\hat{s}_{n+1} \mid \boldsymbol{\beta})$, where $N$ is the number of POIs. We maintain all baseline user embeddings at 256 dimensions and POI embeddings at 128 dimensions. The evaluation metrics include Acc@k and mean reciprocal rank (MRR). The details of the implementation and metric can be found in Appendix B.2.

**Results:** Our brief results are shown in Tab. 1, and consistently surpass all baselines. The comparison with CACSR is particularly noteworthy since it is the latest check-in sequence learning model. We note average performance gains of **17.19%** and **7.49%** over CACSR and ReMVC, respectively. Compared with the SOTA task-specific models, PLSPL and LSTPM realized an average MRR improvement of **9.32%** and **19.88%**. Relative to the time point process models, e.g., NSTPP and DSTPP, our improvements are also pronounced, exceeding **14.86%** and **13.22%**.

## 5.2 Trajectory User Link

**Setups:** Unlike the LP task, the TUL task requires predicting which user generated a given check-in sequence. Therefore, the input information cannot contain any details about the user. We use the

Table 3: Time Prediction (TP) preference results. A lower value indicates better performance. **Red**: the best, Blue: the second best. The units of all metrics are minutes.

| Method | Mobility-LLM | | IFLTTP | | THP | | NSTPP | | DSTPP | | ReMVC | | SML | | CACSR | |
|---|---|---|---|---|---|---|---|---|---|---|---|---|---|---|---|---|
| Datasets | MAE | RMSE | MAE | RMSE | MAE | RMSE | MAE | RMSE | MAE | RMSE | MAE | RMSE | MAE | RMSE | MAE | RMSE |
| Gowalla | 353.89 | 509.55 | 369.82 | 522.83 | 354.6 | 534.52 | 362.39 | 532.99 | 368.76 | 524.33 | 360.62 | 531.46 | 362.03 | 515.16 | 356.37 | 518.22 |
| WeePlace | 28.28 | 35.54 | 29.31 | 36.96 | 29.47 | 36.11 | 29.41 | 37.17 | 28.82 | 37.28 | 29.61 | 36.11 | 28.34 | 36.04 | 28.42 | 35.97 |
| Brightkite | 346.44 | 423.26 | 362.72 | 441.46 | 354.41 | 433.84 | 345.74 | 435.11 | 358.91 | 427.49 | 348.86 | 440.61 | 348.86 | 437.65 | 358.91 | 440.61 |
| FourSquare | 309.78 | 505.03 | 314.74 | 503.01 | 314.52 | 513.11 | 319.39 | 521.69 | 317.84 | 524.22 | 319.39 | 515.13 | 318.46 | 523.71 | 315.67 | 513.11 |

Table 4: Few-shot learning on 5% training data. **Red**: the best, Blue: the second best. We keep the same protocol with the other settings. The results on all training datasets are in Tab. 1 and Tab. 2. The results of few-shot on 1% and 20% can be found in Appendix F.

| Datasets | | Gowalla | | | | WeePlace | | | | Brightkite | | | | FourSquare | | | |
|---|---|---|---|---|---|---|---|---|---|---|---|---|---|---|---|---|---|
| Metric | | Acc@1 | Acc@5 | Acc@20 | MRR | Acc@1 | Acc@5 | Acc@20 | MRR | Acc@1 | Acc@5 | Acc@20 | MRR | Acc@1 | Acc@5 | Acc@20 | MRR |
| Task | Method | | | | | | | | | | | | | | | | |
| LP | DeepMove | 7.58 | 18.01 | 26.29 | 12.58 | 15.47 | 32.89 | 46.85 | 23.35 | 42.25 | 60.56 | 61.72 | 50.62 | 14.44 | 32.26 | 44.34 | 22.73 |
| | LightMove | 7.12 | 16.24 | 23.23 | 11.34 | 15.17 | 32.8 | 47.59 | 23.78 | 41.55 | 57.69 | 59.76 | 49.3 | 12.9 | 28.3 | 39.15 | 20.1 |
| | PLSPL | 8.27 | 18.72 | 26.23 | 13.17 | 14.54 | 31.26 | 46.24 | 23.03 | 43.61 | 59.73 | 61.95 | 51.2 | 16.41 | 35.05 | 46.56 | 25.32 |
| | HMT-LSTM | 7.74 | 17.39 | 25.41 | 12.44 | 14.02 | 30.66 | 45.48 | 22.43 | 41.74 | 57.92 | 58.92 | 49.26 | 13.26 | 29.07 | 40.16 | 20.6 |
| | LSTPM | 7.09 | 16.2 | 23.46 | 11.42 | 15.82 | 31.73 | 45.64 | 24.32 | 36.11 | 49.96 | 52.2 | 42.81 | 14.99 | 33.22 | 45.71 | 23.61 |
| | VaSCL | 8.12 | 17.2 | 25.53 | 12.51 | 14.57 | 32.67 | 47.23 | 24.59 | 42.36 | 60.52 | 61.7 | 50.88 | 14.54 | 31.93 | 44.37 | 22.76 |
| | SimCSE | 5.16 | 12.51 | 18.46 | 8.71 | 11.36 | 27.03 | 41.75 | 19.27 | 39.56 | 58.94 | 61.06 | 48.62 | 14.06 | 32.21 | 45.15 | 22.66 |
| | NSTPP | 7.79 | 18.02 | 25.55 | 12.74 | 13.34 | 28.17 | 42.58 | 20.91 | 38.86 | 53.42 | 55.97 | 46.37 | 15.81 | 34.09 | 45.62 | 24.34 |
| | DSTPP | 7.82 | 17.93 | 25.74 | 12.65 | 14.75 | 31.66 | 46.97 | 23.35 | 41.31 | 57.42 | 58.7 | 49.13 | 12.87 | 28.52 | 39.08 | 20.15 |
| | ReMVC | 7.95 | 17.8 | 25.89 | 12.58 | 15.1 | 32.47 | 47.24 | 24.07 | 42.04 | 58.12 | 60.06 | 50.06 | 13.32 | 29.8 | 40.71 | 20.84 |
| | SML | 7.15 | 16.22 | 23.54 | 11.52 | 13.91 | 29.79 | 44.36 | 22.19 | 39.23 | 53.87 | 56.65 | 46.2 | 14.28 | 31.64 | 44.19 | 22.64 |
| | CACSR | 7.89 | 14.14 | 20.6 | 9.69 | 12.55 | 27.11 | 40.94 | 20.16 | 37.79 | 56.68 | 57.14 | 46.15 | 14.29 | 30.67 | 43.81 | 22.16 |
| | **Mobility-LLM** | 9.98 | 21.82 | 32.02 | 15.74 | 18.27 | 36.59 | 53.23 | 27.15 | 48.49 | 65.31 | 68.66 | 55.64 | 16.86 | 36.47 | 51.36 | 25.98 |
| TUL | TULER | 26.32 | 40.62 | 52.43 | 31.62 | 44.62 | 59.14 | 70.72 | 52.95 | 49.67 | 65.38 | 76.97 | 56.71 | 22.29 | 36.6 | 50.7 | 29.4 |
| | TULVAE | 27.24 | 43.36 | 55.17 | 34.01 | 52.58 | 63.7 | 72.14 | 56.58 | 25.03 | 35.69 | 47.35 | 29.82 | 9.94 | 15.89 | 22.42 | 13.72 |
| | Movesim | 12.28 | 27.73 | 41.55 | 19.17 | 40.61 | 53.33 | 63.86 | 46.16 | 41.1 | 54.05 | 67.93 | 47.21 | 29.9 | 41.88 | 52.98 | 24.69 |
| | S2TUL | 32.13 | 46.84 | 53.43 | 37.93 | 35.88 | 41.55 | 46.56 | 37.16 | 26.01 | 34.32 | 38.15 | 35.96 | 27.61 | 36.3 | 46.74 | 35.5 |
| | VaSCL | 18.95 | 26.76 | 31.46 | 21.59 | 22.19 | 28.26 | 35.44 | 26.42 | 40.75 | 57.2 | 73.66 | 46.6 | 26.29 | 40.4 | 53.66 | 31.45 |
| | SimCSE | 21.2 | 36.02 | 49.89 | 27.15 | 37.89 | 53.88 | 66.82 | 46.28 | 41.45 | 56.68 | 69.7 | 47.88 | 19.7 | 32.78 | 48.66 | 24.89 |
| | ReMVC | 27.76 | 42.29 | 48.14 | 33.63 | 46.08 | 55.73 | 58.49 | 47.89 | 43.38 | 52.84 | 58.61 | 47.55 | 18.85 | 31.63 | 45.11 | 24.8 |
| | SML | 26.78 | 39.45 | 45.19 | 34 | 40.53 | 53.91 | 57.55 | 45.63 | 43.56 | 56.31 | 62.53 | 49.68 | 18.42 | 27.95 | 31.59 | 33.25 |
| | CACSR | 24.67 | 39.52 | 50.7 | 31.36 | 48.85 | 61.46 | 72.04 | 55.73 | 39.83 | 57.24 | 69.76 | 47.51 | 24.95 | 38.3 | 50.28 | 32.37 |
| | **Mobility-LLM** | 55.75 | 69.25 | 76.45 | 61.97 | 66.82 | 77.65 | 83.57 | 71.94 | 69.65 | 79.28 | 85.12 | 74.21 | 51.75 | 64.01 | 73.04 | 57.66 |

corresponding output $\beta$ of HTPP and $\alpha$ with a mean pooling projection head to predict the user $\hat{u}$ who generates this check-in sequence (TUL task). Other settings and evaluation metrics are the same as those used in the LP task.

**Results:** Mobility-LLM consistently surpasses all baselines in Tab. 2, outperforming the all baselines by an average of **47.3**%. Mobility-LLM remains competitive even when compared with the SOTA model, ReMVC, by **17.17**%. Our model performs exceptionally well in the TUL task thanks to its effective extraction of users' travel preferences, allowing for precise identification of users.

### 5.3 Time Prediction

**Setups:** For the time prediction task, we follow the method of IFLTPP [40] using an intensity-free method to model the interaction time as a mixture distribution. We first obtain the mixture weights $w$, means $\mu$ and standard deviations $s$ from the corresponding output $\beta$ of LLM with linear layer. Then we use the TPP projection head built by a mixed distribution function and sample to get the prediction time $\hat{t}_{n+1}$ as follows:

$$p\left(\tau \mid w, \mu, s\right) = \sum_{k=1}^{K} \frac{1}{\tau s_k \sqrt{2\pi}} \exp -\frac{(\log \tau - \mu_k)^2}{2s_k^2}, \hat{t}_{n+1} = \sum_{k=1}^{K} w_k \exp\left(a\mu_k + b + \frac{a^2 s_k}{2}\right), \quad (5)$$

where $k$ represents the number of independent Gaussian distributions in the mixed distribution, $a$ denotes the mean of the whole set and $b$ denotes the standard deviation of the whole set. We sample from the mixture model in the parsing solution. The evaluation metrics include root mean square error (RMSE) and mean absolute error (MAE).

**Results:** Our brief results are shown in Tab. 3. Due to the diversity of user behaviors and the unpredictability of activity timing, even the state-of-the-art (SOTA) models for Temporal Point

Table 5: Ablations on Gowalla dataset in all tasks. Red: the best, Blue: the second best. Our full results can be found in Appendix E.3.1. The setting of **A.1-8** can be found in Appendix E.1, and the settings of **B.1-4** can be found in Appendix E.2.

| Tasks | | LP | | | | TUL | | | | TP | |
|---|---|---|---|---|---|---|---|---|---|---|---|
| Variant | Metic | Acc@1 | Acc@5 | Acc@20 | MRR | Acc@1 | Acc@5 | Acc@20 | MRR | MAE | RMSE |
| **A.1** TinyLlama (Default) | | **11.87** | 25.14 | **36.36** | 18.29 | 80.43 | **86.29** | **88.56** | **83.18** | **353.89** | 509.55 |
| **A.2** TinyLlama-Chat | | 11.59 | 24.33 | 35.76 | 17.88 | 71.83 | 79.37 | 83.38 | 75.38 | 356.34 | 513.76 |
| **A.3** LiteLlama | | 11.56 | **25.39** | 35.78 | **18.41** | 80.53 | 85.11 | 86.12 | 81.11 | 354.71 | **496.36** |
| **A.4** phi-2 | | 11.22 | 24.29 | 35.97 | 17.62 | 72.33 | 79.53 | 83.55 | 75.77 | 358.67 | 512.28 |
| **A.5** pythia-70M | | 11.03 | 24.86 | 35.74 | 17.91 | 79.47 | 85.51 | 86.18 | 81.69 | 356.71 | 513.15 |
| **A.6** pythia-1B | | 11.19 | 24.84 | 35.82 | 18.01 | 78.83 | 84.91 | 88.46 | 80.92 | 354.21 | 511.04 |
| **A.7** pythia-2.8B | | 11.76 | 25.14 | 36.11 | 18.02 | 79.63 | 84.32 | 87.24 | 80.95 | 354.57 | 512.59 |
| **A.8** GPT-2 | | 11.33 | 23.67 | 34.49 | 17.28 | 77.01 | 83.33 | 84.35 | 78.98 | 353.85 | 510.01 |
| **B.1** w/o HTPP | | 11.35 | 24.29 | 33.62 | 17.38 | 72.02 | 79.64 | 80.45 | 75.17 | 361.08 | 517.35 |
| **B.2** w/o VIMN | | 11.01 | 23.27 | 34.03 | 16.83 | 75.88 | 81.79 | 83.72 | 78.06 | 355.65 | 517.26 |
| **B.3** w/o PPEL | | 11.32 | 23.74 | 35.31 | 17.61 | 76.11 | 81.85 | 82.96 | 78.37 | 356.01 | 513.12 |
| **B.4** w/o LLM | | 10.06 | 20.82 | 31.52 | 15.62 | 70.79 | 76.32 | 78.32 | 73.75 | 366.84 | 526.73 |

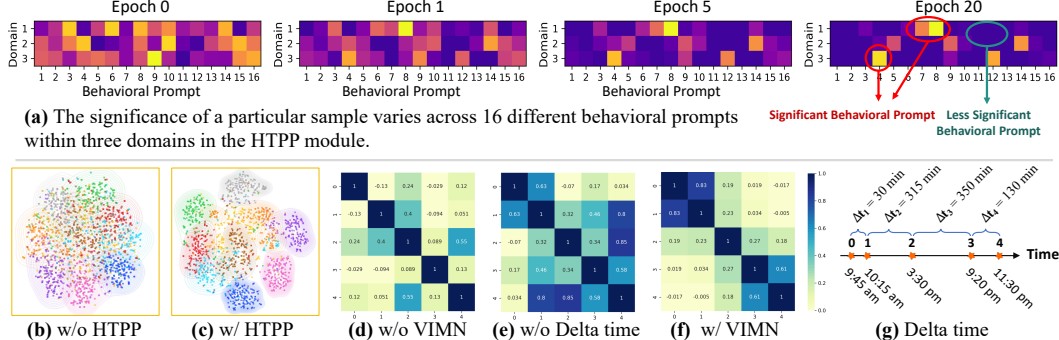

(a) The significance of a particular sample varies across 16 different behavioral prompts within three domains in the HTPP module.

(b) w/o HTPP  (c) w/ HTPP  (d) w/o VIMN  (e) w/o Delta time  (f) w/ VIMN  (g) Delta time

Figure 3: Showcases of the HTPP and VIMN.

Processes (TPP) struggle to achieve excellent results across all datasets. However, our model outperforms all baselines in most cases and does so significantly for most of them.

## 5.4 Few-shot Forecasting

**Setups:** LLMs have recently demonstrated remarkable few-shot learning capabilities [29]. This section assesses whether our reprogrammed LLM retains this ability in different tasks. In our experiments, we maintain consistent splits for training, validation, and test sets in both standard learning (where the full training set is used) and few-shot learning. For few-shot scenarios, we intentionally limit the training data percentage of sample number (i.e., using first 20%, 5%, 1% samples of the training dataset).

**Results:** Our 5% few-shot learning results are in Tab. 4 remarkably excel over all baseline methods, and we attribute this to the successful knowledge activation in our reprogrammed LLM. Interestingly, our approach on both LP and TUL tasks consistently surpasses other competitive baselines, further underscoring the potential prowess of language models as proficient human behavior analysis machines. Concerning recent SOTA models such as NSTPP, DSTPP, S2TUL, ReMVC, and CACSR, our average enhancements surpass **21.4%**, **21.7%**, **86.6%**, **46.2%**, and **45.2%** w.r.t. average on all the metrics. Even with only 5% of the training dataset, our model achieves results comparable to other models using 100% of the training dataset. This is particularly significant for privacy-protected and typically smaller Check-in datasets, as our model can effectively understand the distribution patterns of human behaviors with minimal data.

## 5.5 Model Analysis

**Language Model Variants.** We compare eight representative backbones with varying capacities (**A.1-8** in Tab. 5). We find that the TinyLlama (**A.1** in Tab. 5) backbone model performed the best overall for our task, while its Chat version (**A.2**) is relatively less suitable for reprogramming. Our results indicate that the scaling law is not strictly retained after the LLM reprogramming. Even

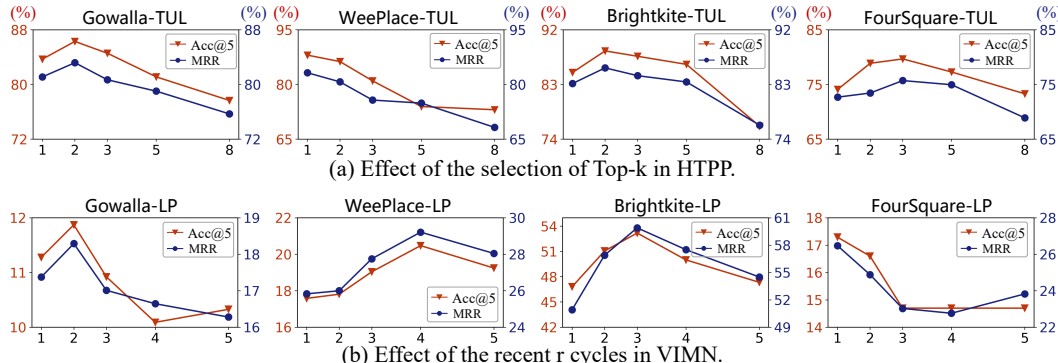

Figure 4: Effects of hyper-parameters validated on different datasets.

the Pythia-70M (**A.5** in Tab. 5) backbone model, which has fewer parameters, can demonstrate advantages in certain datasets and tasks. Therefore, it can be inferred that the suitability of backbone models for different tasks is significantly related to the training strategy of the backbone model, and not just to the number of parameters. For example, in time prediction tasks, GPT-2 (**A.8**) performs better than the 2.7B+ parameter backbone model (**A.4,7**).

**Ablation Study.** As shown in Tab. 5, we find that the removal of (**B.1**) stands as a pivotal element in harnessing the efficiency for the TUL task. We observe a notable average performance degradation of **5.16%** on the TUL task. The inherent habits of human activities can effectively reflect a user's behavioral characteristics, which strongly validates the importance of (**B.1**) in capturing the inherent habits of people's activities. Ablation of this component (**B.2**) results in over **4.09%** degradation on LP task, as people's actions are largely influenced by visiting intentions. Additionally, (**B.3**) allows for more accurate embedding of POI types and semantics, providing a foundation for subsequent modules and tasks. In the w/o LLM setting (**B.4**), we replace the large language model with a standard transformer. (**B.4**) also shows that even without using an LLM as the base model, our reprogramming still achieves excellent results, even surpassing many baselines.

**Reprogramming Interpretation.** We provide three case studies to visualize the improvements brought by HTPP and VIMN. In Fig. 3a, we visualize the significance of different prompt words in three domain pools within the HTPP model. **Orange** indicates high significance, and **Purple** indicates low significance. It can be observed that as training progresses, the prompt words for this sample in each domain concentrate on a few words. This also indicates that user behavior is diverse and personalized. To more intuitively see the effect of HTPP, we select 10 different users, each with several samples. We visualize the output of LLM with t-SNE [43] corresponding to the samples generated by these 10 users in Fig. 3b and Fig. 3c. It can be seen that with the addition of HTPP, the model can more comprehensively capture user behavior patterns and better recognize the behavioral characteristics of each user. Similarly, to better visualize the effect of VIMN, we select a continuous sequence of five user behavior points. Fig. 3d-f show the Pearson correlation coefficients between the five outputs $h$ without using the VIMN, without considering Delta Time as input, and when using VIMN, respectively. Fig. 3g shows the time intervals among these five points. It can be observed that with the use of the VIMN, the closer the time, the greater the influence on the current state, which aligns with human behavior patterns. Since plans cannot keep up with changes, people's decisions are mostly influenced by recent behaviors. Therefore, our model uses Delta Time as a constraint for correlation, effectively mimicking human activity patterns and achieving better prediction results.

**Hyperparameter Analysis.** There is a significant difference in user behavior patterns across different datasets. As shown in Fig. 4a, in larger datasets like Foursquare, user behavior is more diverse, and larger K values yield better results. In Fig. 4b, it can also be seen that the number of nearby cycles in VIMN is highly correlated with the distribution of time intervals in the dataset.

**Reprogramming Efficiency.** Tab. 6 provides an overall efficiency analysis with and without the backbone LLM. Our proposed reprogramming network itself is lightweight in activating the LLM's ability for different tasks (i.e., fewer than 10 million trainable parameters; only around **3.4%** of the total parameters in TinyLlama), and the leveraged backbones actually cap the overall efficiency. This is favorable even compared to the parameter-efficient fine-tuning methods (e.g., QLoRA [8])

Table 6: Efficiency analysis of Mobility-LLM on WeePlace dataset on all tasks. Param. represents the total parameters of the model. Mem. denotes the GPU Memory. The ratio represents the ratio of trainable parameters (including the trainable parameters in QLoRA and the reprogramming parameters). The Time column denotes the total training time.

| Tasks | | LP | | | | TUL | | | | TP | | |
|---|---|---|---|---|---|---|---|---|---|---|---|---|
| Variant | Metric | Param. | Mem. | Ratio | Time | Param. | Mem. | Ratio | Time | Param. | Mem. | Ratio | Time |
| TinyLlama (Default) | | 1.07B | 11.3GB | 3.72% | 5.45h | 1.03B | 10.6GB | 3.34% | 13.24h | 1.03B | 10.6GB | 3.42% | 2.76h |
| TinyLlama-Chat | | 1.07B | 11.3GB | 3.72% | 5.45h | 1.03B | 10.6GB | 3.34% | 13.24h | 1.03B | 10.6GB | 3.41% | 2.76h |
| LiteLlama | | 434M | 8.7GB | 5.42% | 4.58h | 412M | 8.2GB | 5.02% | 5.65h | 414M | 8.2GB | 4.60% | 2.36h |
| phi-2 | | 2.82B | 21.7GB | 1.12% | 8.95h | 2.81B | 20.76GB | 1.03% | 17.59h | 2.81B | 20.76GB | 1.21% | 6.42h |
| pythia-70M | | 63M | 2.74GB | 27.23% | 2.76h | 53M | 2.56GB | 20.21% | 4.92h | 51M | 2.56GB | 22.03% | 1.13h |
| pythia-1B | | 1.08B | 12.03GB | 3.89% | 3.89h | 1.01B | 11.87GB | 3.21% | 6.5h | 1.01B | 11.87GB | 3.43% | 3.98h |
| pythia-2.8B | | 2.73B | 22.4GB | 1.19% | 11.7h | 2.71B | 21.76GB | 1.23% | 11.7h | 2.71B | 21.76GB | 1.15% | 6.53h |
| GPT-2 | | 0.138B | 3.67GB | 19.32% | 3.06h | 0.135B | 3.23GB | 19.11% | 4.51h | 0.135B | 3.23GB | 19.04% | 1.27h |
| w/o LLM | | 0.012B | 3.23GB | 92.23% | 2.42h | 0.011B | 3.12GB | 94.32% | 4.95h | 0.011B | 3.12GB | 97.53% | 1.2h |

in balancing task performance and efficiency. In terms of runtime, the total training time with the advanced fine-tuning framework is acceptable compared to not using LLMs.

# 6   Conclusion

In conclusion, our work presents **Mobility-LLM**, a unified framework leveraging large language models (LLMs) to analyze check-in sequences and understand human mobility behaviors. By incorporating the visiting intention memory network (VIMN) and the human travel preference prompts (HTPP), our model excels in various tasks. Moreover, our model exhibits robust few-shot learning capabilities, outperforming existing methods by an average of 23.6% to 38.3%. Our work paves the way for a more comprehensive and accurate analysis of human mobility, benefiting individuals, businesses, and urban management.

**Limitations** The sets of POIs in different datasets (which usually cover different regions) are unique. Therefore, our proposed model is trained on one dataset, its learned information about the set of POIs is not easily transferable to another dataset. Different sets of POIs have different functionalities and usually have a different number of POIs, making many modules (such as embedding and predictor) technically untransferable in a zero-shot setting. Future work will focus on developing universal user and POI embeddings to enhance cross-dataset migration and improve model versatility.

**Acknowledgment.** This work was supported by the National Natural Science Foundation of China (No. 62372031) and the Beijing Natural Science Foundation (Grant No. 4242029).

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

# Appendix

Here we introduce the related work of this paper in Sec.A. The implement details including experiment settings and evaluation metrics can be found in Sec.B. The datasets and Partially Frozen Attention (PFA) LLM settings are shown in Sec. C. An introduction to Baselines is presented in Sec.D. We display the results of ablation and few shot on all datasets in Sec.E and Sec.F respectively. Moreover, We introduce the calculation process of data in feed-forward layer and geohash embedding layer in Sec.G and Sec.H in detail respectively. We also list POI categories in Sec.I.

## A  Related Works

### A.1  Mobility Data Mining

Location-based services have given rise to a new and promising research topic known as mobility data mining, which has led to the emergence of three significant tasks that contribute to enhancing the quality of services: next location prediction (LP), next time prediction (TP), and trajectory user link (TUL). Recent studies have confirmed that deep learning techniques, specifically recurrent neural networks (RNNs) and attention mechanisms, are highly effective in capturing sequential and periodic patterns of human mobility. By combining deep learning techniques, researchers have made significant advancements in capturing both the sequential and periodic patterns of human mobility. The core of these models is the modeling of check-in sequences, which leads to improved accuracy in location prediction and trajectory analysis.

LP aims to anticipate a user's future location based on their historical movement. Several notable models have emerged as prominent approaches in LP. DeepMove [10] leverages RNNs and attention mechanisms to capture the spatial-temporal intentions in users' location data and predict their next destination. STAN [33] introduces a spatial-temporal attention network that incorporates spatial and temporal contexts for accurate prediction. LSTPM [59] focuses on long and short-term patterns in user trajectory using an attention-based LSTM [20] model. SERM [52] utilizes an encoder-decoder architecture with a spatial-temporal residual network to capture user preferences and predict future locations. PLSPL [47] trains two LSTM models for location- and category-based sequences to capture the user's preference. LightMove [22] designs neural ordinary differential equations to enhance robustness against sparse or incorrect inputs. HMT-GRN [25] alleviates the data sparsity problem by learning different User-Region matrices of lower sparsities in a multitask setting. Graph-Flashback [38] constructs a spatial-temporal knowledge graph to enhance the representation of POIs, having a great advantage when dealing with the historical sequence input of the same length. GETNext [51] introduces a user-agnostic global trajectory flow map as a means to leverage the abundant collaborative signals.

TUL is a significant task that focuses on establishing connections between different trajectories, facilitating the analysis of user movement patterns, and uncovering valuable insights about their behavior. Notable models have been specifically developed to address the challenge of predicting trajectory links. TULER [13] takes advantage of advanced algorithms to establish links between trajectories, allowing for a comprehensive understanding of user movement patterns. TULVAE [62] uses latent variables to model the variability in trajectories, capturing hierarchical and structural semantics and improving the identification and linking performance. MoveSim [11] simulates human mobility using a generative adversarial framework that incorporates attention mechanisms to capture complex spatial-temporal transitions in human mobility. DeepTUL [34] utilizes deep learning techniques to extract representations from trajectory data and facilitate the prediction of trajectory links. S2TUL [7] utilizes graph convolutional networks and sequential neural networks to capture trajectory relationships and intra-trajectory information. GNNTUL [60] employs graph neural networks for human mobility and associates the traces with users on social networks.

TP focuses on estimating the time at which a user is likely to visit their next location. To accomplish this, it is common practice to use intensity functions to represent the rate or density of event occurrences, various models have been developed to model the intensity function and make accurate time predictions effectively. Modeling the intensity function using RNNs or attention mechanisms is a common approach for predicting the occurrence of events. IFLTPP [40] approximates any distribution of inter-event times using normalizing flows and mixture distributions. RMTPP [9] utilizes RNNs to model the intensity function. SAHP [57] combines the Hawkes process with self-

attention mechanisms to capture the temporal dependencies and spatial influences in event sequences. THP [64] combines the Hawkes process with transformer-based architectures to capture temporal dependencies in event sequences. NSTPP [17] utilizes neural ODEs to model discrete events in continuous time and space, enabling the learning of complex distributions in spatial and temporal domains. IMTPP [18] models the generative processes of observed and missing events and utilizes unsupervised modeling and inference methods for time prediction. DSTPP [53] proposes a novel parameterization framework that uses diffusion models to learn complex joint distributions.

In recent years, many sequence representation models have become a hot topic, including those for natural language sequences and spatial-temporal trajectory sequences. For natural language sequences, SimCSE [14] is a classical sequence representation language model that utilizes a straightforward contrastive learning framework. It employs standard dropout as noise, making it easy to implement. However, it cannot effectively distinguish between hard negative samples or semantically similar sequences. Thus, VaSCL [54] leverages neighborhood information to generate virtual augmentations, improving the quality of data transformations compared to traditional methods. Inspired by models for natural language sequence representation, researchers in the spatial-temporal domain have also conducted similar studies on spatial-temporal sequences. SML [61] first employs self-supervised contrastive learning to effectively manage sparse and noisy trajectory data, enhancing trajectory representation through spatial-temporal data augmentation. However, the spatial-temporal augmentation in SML needs to be manually specified, which is relatively cumbersome. Therefore, ReMVC [55] effectively addresses this issue by adopting a cross-view contrastive approach. It uses contrastive learning to extract meaningful region representations, improving intra-view and inter-view learning. And CACSR [15] effectively adds adversarial perturbations and automated data augmentation, enhancing contrastive training processing. While it lacks interpretability and does not understand the semantic information of time and space from the actual semantics of human activities.

In summary, it can be seen that whether it is end-to-end models designed for specific tasks or spatiotemporal sequence representation learning models, their understanding of human activities is not yet deep enough, and they lack a profound understanding of the essence and spatial-temporal patterns behind human activities.

### A.2 Cross-domain Application of LLMs

We have witnessed the great success of Large Language Models (LLMs) in natural language processing [37], and some cross-domain (such as computer vision [36], time series, and graph-related tasks. In the field of graph theory, To tackle graph-related tasks, Graph Neural Networks (GNNs) have emerged as one of the most popular choices for processing and analyzing graph data. While GNNs excel at capturing structural information, their reliance on semantically constrained embeddings as node features limits their ability to fully express the complexities of the nodes. By incorporating LLMs, GNNs can be augmented with stronger node features that effectively capture both structural and contextual aspects. TAPE utilizes semantic knowledge that is pertinent to the nodes (e.g., papers) generated by LLMs to enhance the quality of initial node embeddings in GNNs. Furthermore, InstructGLM replaces the predictor in GNNs with LLMs, harnessing the expressive capabilities of natural language through techniques like graph flattening and prompt-based instruction design. MoleculeSTM aligns GNNs and LLMs within a shared vector space, integrating textual knowledge into graphs (e.g., molecules) to enhance reasoning capabilities.

In the field of Computer Vision (CV), ViT is an image classification model based on the Transformer architecture, which has achieved excellent performance on multiple benchmark image classification datasets. CLIP is a large-scale pre-trained model developed by OpenAI that aligns images and text, enabling simultaneous learning of representations for both modalities, which allows for interactive understanding between images and text. DALL-E utilizes a large-scale pre-trained language model to transform text into images through a generative approach, showcasing remarkable capabilities in image generation. TEST [41], as a time series (TS) embedding method tailored for Large Language Models (LLMs), generates embeddings that capture similarity, instance-wise, feature-wise, and text-prototype alignment for TS tokens. Time-LLM [24] introduces a reprogramming framework. Given the challenge of limited training data, recent studies turn to adopting large language models (LLMs) to address cross-domain applications. In the field of time series analysis, LLM4TS [4] is the pioneering method that aligns pre-trained Large Language Models with temporal characteristics, introducing a two-level aggregation method to effectively incorporate multi-scale temporal data

into pre-trained LLMs. One-Fits-All [63] is a unified framework that leverages a frozen pre-trained language model to attain state-of-the-art or comparable performance across various major types of time series analysis tasks. AutoTimes [30] facilitates the tokenization of time series into the embedding space of LLMs and intelligently utilizes the inherent token transitions to effectively predict time series in an autoregressive manner. In the field of computer vision, LM4VE [35] incorporates a frozen transformer block from an LLM as a general-purpose visual encoder layer. To tackle graph-related tasks, TAPE [19] utilizes semantic knowledge that is pertinent to the nodes generated by LLMs to enhance the quality of initial node embeddings in GNNs. MoleculeSTM [27] aligns GNNs and LLMs within a shared vector space, integrating textual knowledge into graphs to enhance reasoning capabilities.

GPT4TS [42] utilizes PLMs by freezing the self-attention feed-forward layers. For visual encoding tasks, LM4VisualEncoding [35] incorporates a frozen transformer block from a PLM as a general-purpose visual encoder layer. RLMRec [39] integrates the semantic space of PLMs and collaborative relational signals using an alignment framework. An LLM agent framework is proposed for flexible and efficient personal mobility generation [46].

While these studies provide valuable insights, it is important to recognize that their methods cannot be directly applied to the domain of trajectory learning. Trajectory data exhibits distinct spatial-temporal characteristics and contains unique information that requires customized approaches and considerations.

## B  Implement Details

### B.1  Settings

The Mobility-LLM model was constructed using the PyTorch[5]. The loss function is a cross-entropy loss for the LP and TUL tasks and an MAE loss for the TP task. The performance on the validation sets determines the hyper-parameters and the best models. All experiments are performed five times, and the means and standard deviations are calculated. To make a fair comparison, for all methods, the embedding dimension $d$ is 256, while the hidden state $h$ has a size of 256. The learning rate is 0.001. The model is pre-trained for 100 epochs on the training sets with the early-stopping mechanism of 10 patients. All trials have been conducted on Intel Xeon E5-2620 CPUs and NVIDIA RTX A40 GPUs.

### B.2  Evaluation Metrics

Four metrics are used for evaluating the models: mean absolute error (MAE), root mean squared error (RMSE), Accuracy at K (ACC@K), and Mean Reciprocal Rank (MRR). MAE and RMSE quantify absolute errors. ACC@K measures the accuracy of the predictions within the top K ranks. MRR calculates the average of the reciprocal ranks, where a higher rank indicates a better performance.

$$\text{MAE} = \frac{1}{m}\sum_{i=1}^{m}\left|\widehat{\mathbf{Y}}_i - \mathbf{Y}_i\right|, \text{RMSE} = \sqrt{\frac{1}{m}\sum_{i=1}^{m}\left(\widehat{\mathbf{Y}}_i - \mathbf{Y}_i\right)^2}, \tag{6}$$

where $m$ is the number of predicted values, $\widehat{\mathbf{Y}}_i$ is the predicted value, $\mathbf{Y}_i$ is the target value.

$$\text{ACC@K} = \frac{1}{m}\sum_{i=1}^{K}\left(\widehat{\mathbf{Y}}_i == \mathbf{Y}_i\right), \text{MRR} = \frac{1}{m}\sum_{i=1}^{m}\frac{1}{\text{rank}_i}, \tag{7}$$

where $m$ is the total number of queries, $\text{rank}_i$ is the rank position of the first relevant result for the $i - th$ query.

## C  Check-in Datasets

### C.1  Datasets Introduction

**Gowalla** was a location-based social networking service where users shared their locations by checking-in. The dataset includes 6,442,890 check-ins made by 196,591 users from February 2009 to

---

[5]HTPPs://pytorch.org

October 2010. Additionally, it contains the undirected friendship network, which consists of 950,327 edges. This dataset provides valuable information for studying mobility patterns and social network analysis.

- **Nodes:** 196,591
- **Edges:** 950,327
- **Check-ins:** 6,442,890
- **Time Period:** Feb 2009 - Oct 2010
- **Data Fields:** User ID, Check-in Time, Latitude, Longitude, Location ID

**Weeplaces** dataset was collected from a service that visualizes users' check-in activities across multiple location-based social networks, including Facebook Places, Foursquare, and Gowalla. It consists of 7,658,368 check-ins generated by 15,799 users over 971,309 locations. The data primarily captures the check-in history and friend connections of users.

- **Check-ins:** 7,658,368
- **Users:** 15,799
- **Locations:** 971,309
- **Data Fields:** User ID, Check-in Time, Latitude, Longitude, Location Name, Category

**Brightkite** was a location-based social networking service where users could check in at various locations and share their locations with friends. The dataset includes 4,491,143 check-ins made by 58,228 users from April 2008 to October 2010. This dataset is useful for researching user mobility and social interaction patterns.

- **Check-ins:** 4,491,143
- **Users:** 58,228
- **Time Period:** Apr 2008 - Oct 2010
- **Data Fields:** User ID, Check-in Time, Latitude, Longitude, Location ID

**Foursquare** is a popular location-based social networking service where users check in at various venues. The dataset includes check-in data from multiple cities, such as New York City and Tokyo, over a period from April 2012 to February 2013. It contains 227,428 check-ins in New York City and 573,703 check-ins in Tokyo. Each entry provides a user ID, check-in time, latitude, longitude, and venue category.

- **Check-ins in NYC:** 227,428
- **Check-ins in Tokyo:** 573,703
- **Time Period:** Apr 2012 - Feb 2013
- **Data Fields:** User ID, Check-in Time, Latitude, Longitude, Venue Category

### C.2 The Statics of Processed Datasets

We can see that the dataset we selected includes the Gowalla dataset, which has a large number of samples, users, and Points of Interest (POI), as well as the Foursquare dataset, which has a small number of users but a large number of POIs. Additionally, we have the Brightkite and WeePlace datasets, which have a small number of both users and POIs. These datasets cover different scenarios and can fully validate the model's comprehensive capabilities.

### C.3 Partially Frozen Attention (PFA) LLM

The frozen pre-trained transformer (FPT) has demonstrated effectiveness in a variety of downstream tasks across non-language modalities [32]. We adopt a partially frozen attention (PFA) LLM, specifically designed to enhance prediction accuracy in check-in sequences prediction.

The difference between the FPT and our PFA primarily lies in the configuration of frozen attention layers. In the FPT framework, both the multi-head attention and feed-forward layers are frozen during

Table 7: The statics of Processed Datasets

| Datasets | #Samples | #Users | #POIs |
|---|---|---|---|
| Gowalla | 413,563 | 5,853 | 52,032 |
| WeePlace | 104,762 | 1,028 | 9,295 |
| Brightkite | 44,716 | 431 | 3,554 |
| FourSquare | 60,734 | 703 | 11,117 |

training, as these layers contain the most significant portion of the learned knowledge within the LLM. In the PFA, we maintain the first $F$ layers identical to the FPT, but crucially, we unfreeze the last $U$ multi-head attention layers since the attentions offer effective handling of spatial-temporal dependencies in data. Consequently, our Mobility-LLM can adapt to check-in datasets while preserving the foundational knowledge acquired during pre-training.

# D   Overview of Baselines

**DeepMove** [10] is an attentional recurrent network designed for predicting human mobility from lengthy and sparse trajectories. It introduces a multi-modal embedding recurrent neural network that captures complex sequential transitions by embedding multiple factors influencing human mobility. DeepMove further incorporates a historical attention model with dual mechanisms to effectively capture multi-level periodicity. This historical attention model enhances the recurrent neural network's ability to predict mobility patterns by leveraging the inherent periodic nature of human movement.

**LightMove** [22] is a lightweight and accurate deep learning-based method developed for predicting the next locations of taxicabs in order to enhance targeted advertising on taxicab rooftop devices. The paper focuses on Motov, a leading company in South Korea's taxicab rooftop advertising market, and aims to leverage demographic information of locations to improve the preparation of targeted advertising campaigns.

**PLSPL** [47] addresses the task of recommending the next Point of Interest (POI) for users based on their historical check-in data. The main objective is to capture both the users' general taste and their recent sequential behaviors, as these factors are crucial in making accurate recommendations. However, existing methods often assume the same dependencies for all users, disregarding the fact that different users may have varying preferences and dependencies on these two aspects.

**HMT-LSTM** [25] addresses the challenging task of predicting the next Point-of-Interest (POI) that a user is likely to visit in personalized recommender systems. One of the main difficulties in this task is the large search space of possible POIs in the region, which leads to data sparsity issues in existing works and hampers performance.

**LSTPM** [59] proposes a method for POI recommendation by modeling long-term and short-term preferences. It uses a nonlocal network for capturing long-term dependencies and a geo-dilated RNN for considering geographical relationships. This approach improves the reliability of recommendation results compared to existing methods.

**TULER** [13] addresses the task of understanding human trajectory patterns in Location-Based Social Networks (LBSNs) applications. This task is crucial for various applications like personalized recommendation and preference-based route planning. Existing methods often classify trajectories or their segments into predefined categories based on spatial-temporal values and activities, such as walking or jogging. However, the paper focuses on a novel problem called Trajectory-User Linking (TUL), which aims to identify and link trajectories to the users who generated them in LBSNs.

**TULVAE** [62] addresses the important task of Trajectory-User Linking (TUL) in Geo-tagged social media (GTSM) applications. It utilizes a neural generative architecture with stochastic latent variables that span hidden states in an RNN. By incorporating variational autoencoder techniques, TULVAE can capture the hierarchical and structural semantics of trajectories using high-dimensional latent variables. Moreover, TULVAE addresses the data sparsity challenge by leveraging large-scale unlabeled data, providing more comprehensive representations of user mobility patterns.

**MoveSim** [11] is a framework for the realistic simulation of massive human mobility data. Existing solutions for mobility simulation have limitations in generating high-quality data due to complex transitions and intricate regularities in human mobility patterns.

**S2TUL** [7] is a flexible Semi-Supervised framework for Trajectory-User Linking (TUL) with five main components.

**IFLTPP** [40] is a novel approach that models event sequences with irregular intervals using temporal point processes. Unlike the standard intensity-based methods, IFLTPP directly models the conditional distribution of inter-event times. Drawing inspiration from normalizing flows, IFLTPP captures complex dependencies in the data while maintaining practicality and interpretability through a simple mixture model.

**THP** [64] is a model designed for handling massive event sequence data with complex temporal dependencies. To address the limitations of existing recurrent neural network-based models, THP utilizes the self-attention mechanism of Transformers, which effectively captures short-term and long-term dependencies in the data.

**NSTPP** [17] is an approach that addresses the retrieval problem of continuous-time event sequences (CTESs) using marked temporal point processes (MTPP). It improves the retrieval performance by applying a trainable unwarping function to the query sequence, making it comparable to the corpus sequences.

**DSTPP** [53] is a framework designed for modeling spatial-temporal point processes (STPPs) by leveraging diffusion models. It addresses the limitations of existing solutions that assume conditional independence between time and space.

**ReMVC** [55] is a model designed for unsupervised region representation learning from unlabeled urban data. It addresses the limitations of previous methods by leveraging contrastive learning for multi-view region representation learning.

**VaSCL** [54] is a framework designed for data augmentation in contrastive representation learning, with a focus on the challenging context of natural language processing (NLP). Unlike other domains, NLP lacks general rules for effective data augmentation due to the discrete nature of language.

**SML** [61] is a framework that tackles the challenges of extracting meaningful supervised signals from sparse and noisy human mobility data obtained from location-based services (LBS). The framework aims to leverage this data for various applications, including location recommendation, anomaly trajectory detection, crime discrimination, and epidemic tracing.

**CASCR** [15] is a model that addresses the accurate representation learning of user-generated check-in sequences in human mobility data. It introduces a contrastive pre-training approach specifically designed for check-in sequence representation learning, eliminating the need for manual adjustments to data augmentation strategies.

## E  Variants

### E.1  Variants of the Used LLMs

**TinyLlama [6].** TinyLlama-1.1B is a lightweight language model developed by a team of researchers at the Singapore University of Technology and Design (SUTD). It has 1.1 billion parameters and was pre-trained on about 3 trillion tokens. The training process for TinyLlama was done in 90 days using 16 A100-40G GPUs. The researchers explored the model's performance when the number of tokens suggested by the scaling law was exceeded by training the mini-model using a large amount of data. The model also used various optimizations such as flash attention 2, FSDP (Fully Sharded Data Parallel), and xFormers to improve the efficiency and throughput of training. The application of these techniques gives TinyLlama a significant advantage in terms of training speed and memory usage.

TinyLlama adopts exactly the same architecture and tokenizer as Llama 2. This means TinyLlama can be seamlessly integrated into many open-source projects built upon Llama. Additionally, TinyLlama is compact with only 1.1B parameters, allowing it to cater to a multitude of applications demanding a restricted computation and memory footprint.

**TinyLlama-Chat [6].** The TinyLlama-Chat model is fine-tuned on top of TinyLlama/TinyLlama-1.1B-intermediate-step-1431k-3T. Following HF's Zephyr's training recipe, the model was "initially fine-tuned on a variant of the UltraChat dataset, which contains a diverse range of synthetic dialogues generated by ChatGPT. We then further aligned the model with TRL's DPOTrainer on the openbmb/UltraFeedback dataset, which contains 64k prompts and model completions that are ranked by GPT-4.".

**LiteLlama [12].** LiteLlama Series Models Published by Xiaotian Han, Texas A&M University. The LiteLlama series of models have a wide range of applications on edge devices, such as smartphones, IoT devices, and embedded systems, which usually have limited computational power and storage space, and they are unable to run large language models efficiently. Therefore, it is particularly important to explore small models in depth.

The main models in the LiteLlama family are LiteLlama-460M. LiteLlama presents an open-source reproduction of Meta AI's LLaMa 2. It has 460M parameters with 1T tokens. LiteLlama-460M-1T was trained on the RedPajama dataset and the text was tokenized using GPT2Tokenizer. The authors evaluated the model on the MMLU task, comparing the parameter sizes to those of a large model of equal capacity, and the results demonstrate that with a significantly reduced number of parameters, LiteLlama-460M-1T still achieves results comparable to or better than the other models.

**Phi-2 [1].** The Phi family of language models are small language models introduced by Microsoft. The goal of the Phi family of language models is to demonstrate that by constructing high-quality pre-training data, small models can achieve significantly better performance than larger models at lower training costs. The Phi family of language models includes Phi-1, Phi-1.5, and Phi-2.

Phi-2 is a Transformer with 2.7 billion parameters. Trained using the same data sources as Phi-1.5, Phi-2 incorporates a new data source consisting of various NLP synthetic texts and filtered websites (for safety and educational value). It showcases a nearly state-of-the-art performance among models with less than 13 billion parameters, especially when assessed against benchmarks testing common sense, language understanding, and logical reasoning.

**Pythia-70M, 1B, 2.8B [2].** The Pythia Scaling Suite is a collection of models developed to facilitate interpretability research. It contains two sets of eight models of sizes 70M, 160M, 410M, 1B, 1.4B, 2.8B, 6.9B, and 12B. For each size, there are two models: one trained on the Pile, and one trained on the Pile after the dataset has been globally deduplicated. All 8 model sizes are trained on the exact same data, in the exact same order. Additionally, 154 intermediate checkpoints per model are hosted on Hugging Face as branches.

The Pythia model suite was deliberately designed to promote scientific research on large language models, especially interpretability research. Despite not centring downstream performance as a design goal, the models match or exceed the performance of similar and same-sized models, such as those in the OPT and GPT-Neo suites.

**GPT-2 [37].** GPT-2 is a transformers model pre-trained on a very large corpus of English data in a self-supervised fashion. This means it was pre-trained on the raw texts only, with no humans labeling them in any way. More precisely, it was trained to guess the next word in sentences.

Inputs are sequences of continuous text of a certain length, and the targets are the same sequence, shifted one token to the right. The model uses internally a masking mechanism to make sure the predictions for the $i$-th token only use the inputs from 1 to $i$ but not the future tokens.

### E.2 Variants of Ablation

To further evaluate the effects of different components in Mobility-LLM, we conduct ablation experiments and analyze experimental results on all datasets. We compare these four variants on three downstream tasks.

- **w/o HTPP:** We remove the HTPP Module. The rest of the settings are the same as Mobility-LLM. We use this setting to evaluate the function of the HTPP Module.
- **w/o VIMN:** We use full-connection to replace the VIMN module. The rest of the settings are the same as Mobility-LLM. We use this setting to evaluate the function of the VIMN module.
- **w/o PPEL:** We use a learnable parameter to represent a POI. We use this setting to evaluate the function of the PPEL.

- **w/o LLM:** We use a Transformer layer to replace the LLM. The rest of the settings are the same as Mobility-LLM. We use this setting to evaluate the function of the LLM.

### E.3 Rusults of Variants on Different Datasets

### E.3.1 Rusults of Variants on WeePlace

Table 8: Ablations on WeePlace dataset in all tasks. Red: the best, Blue: the second-best.

| Tasks | | LP | | | | TUL | | | | TP | |
|---|---|---|---|---|---|---|---|---|---|---|---|
| Variant | Metic | Acc@1 | Acc@5 | Acc@20 | MRR | Acc@1 | Acc@5 | Acc@20 | MRR | MAE | RMSE |
| **TinyLlama (default)** | | 20.47 | 39.22 | 56.69 | 29.21 | 79.03 | 88.04 | 91.48 | 83.21 | 28.28 | 35.54 |
| **TinyLlama-Chat** | | 19.93 | 38.52 | 56.20 | 28.32 | 68.69 | 80.90 | 86.21 | 72.25 | 28.61 | 36.12 |
| **LiteLlama** | | 20.06 | 40.20 | 55.28 | 29.67 | 76.20 | 86.49 | 87.01 | 81.30 | 28.12 | 34.45 |
| **phi-2** | | 19.10 | 38.23 | 55.97 | 28.51 | 70.72 | 79.03 | 83.38 | 75.87 | 28.09 | 34.87 |
| **pythia-70M** | | 18.70 | 38.59 | 54.16 | 28.57 | 76.05 | 86.98 | 90.08 | 78.78 | 28.76 | 34.90 |
| **pythia-1B** | | 19.11 | 37.78 | 53.89 | 28.45 | 76.22 | 86.38 | 90.74 | 81.44 | 28.68 | 35.15 |
| **pythia-2.8B** | | 19.77 | 39.77 | 54.33 | 27.77 | 75.28 | 82.42 | 89.57 | 80.49 | 28.70 | 36.36 |
| **GPT-2** | | 19.11 | 36.03 | 54.47 | 27.59 | 73.93 | 83.15 | 86.87 | 75.94 | 27.88 | 34.97 |
| **w/o HTPP** | | 19.32 | 38.31 | 51.42 | 26.89 | 76.73 | 83.89 | 84.94 | 78.25 | 28.85 | 35.43 |
| **w/o VIMN** | | 19.72 | 36.61 | 53.69 | 27.05 | 74.77 | 86.08 | 86.28 | 79.56 | 28.22 | 35.43 |
| **w/o PPE** | | 19.19 | 37.67 | 55.00 | 27.53 | 77.21 | 84.96 | 88.29 | 81.85 | 28.82 | 34.76 |
| **w/o LLM** | | 17.66 | 32.84 | 49.19 | 24.80 | 68.15 | 81.34 | 83.52 | 74.59 | 28.53 | 35.68 |

**Language Model Variants.** We compared eight representative backbones on WeePlace dataset with varying capacities in Tab. 8. We found that the TinyLlama backbone model performed the best overall for our task. In particular, TinyLlama achieved the highest scores in LP (Acc@1, Acc@20) and TUL (Acc@1, Acc@5, Acc@20, MRR) tasks. The Chat version of TinyLlama was relatively less suitable, showing lower performance across most metrics. Interestingly, LiteLlama exhibited competitive performance, achieving the best Acc@5 and MRR in the LP task. Despite having fewer parameters, Pythia-70M demonstrated advantages in some tasks, achieving the second-best Acc@5 in the TUL task and the best MAE in the TP task. This suggests that the suitability of backbone models for different tasks is significantly influenced by the training strategy rather than just the number of parameters. For example, in trajectory prediction tasks, GPT-2 performed better than some models with larger parameter sizes such as Pythia-2.8B.

**Ablation Study.** As shown in Tab. 8, we conducted an ablation study to understand the impact of different components on performance. Removing HTPP (**w/o HTPP**) led to a significant decrease in performance, particularly in the LP task where Acc@20 decreased of **9.29%** and MRR of **7.94%**. This highlights the importance of HTPP in the model's ability to make accurate predictions. Similarly, removing VIMN (**w/o VIMN**) resulted in noticeable performance drops in both LP and TUL tasks, indicating its role in maintaining robust prediction capabilities. Without PPE (**w/o PPE**), the model showed a moderate decrease in performance, with MRR in the LP task decreased of **5.75%**. The removal of LLM (**w/o LLM**) had the most pronounced negative effect across all tasks, with LP Acc@1 decreased of **13.7%** and MRR of **15.1%**, and TUL metrics significantly reduced. This underscores the crucial role of the large language model in the overall framework, as even replacing it with a standard transformer could not achieve comparable performance.

Table 9: Ablations on Brightkite dataset in all tasks. Red: the best, Blue: the second-best.

| Tasks | | LP | | | | TUL | | | | TP | |
|---|---|---|---|---|---|---|---|---|---|---|---|
| Variant | Metic | Acc@1 | Acc@5 | Acc@20 | MRR | Acc@1 | Acc@5 | Acc@20 | MRR | MAE | RMSE |
| **TinyLlama (default)** | | 53.18 | 68.31 | 74.11 | 59.89 | 83.06 | 88.52 | 90.35 | 85.73 | 346.44 | 423.26 |
| **TinyLlama-Chat** | | 52.03 | 66.31 | 72.81 | 58.31 | 74.33 | 81.67 | 85.07 | 77.54 | 347.79 | 425.90 |
| **LiteLlama** | | 51.74 | 69.13 | 72.93 | 59.98 | 83.08 | 87.40 | 87.51 | 83.18 | 345.51 | 411.89 |
| **phi-2** | | 50.17 | 65.87 | 73.02 | 57.52 | 74.40 | 81.59 | 85.24 | 77.86 | 351.12 | 423.40 |
| **pythia-70M** | | 49.22 | 67.48 | 72.70 | 58.82 | 81.66 | 87.63 | 87.92 | 84.19 | 348.50 | 427.10 |
| **pythia-1B** | | 50.23 | 67.70 | 73.23 | 59.03 | 81.65 | 86.84 | 89.89 | 83.40 | 346.06 | 424.50 |
| **pythia-2.8B** | | 52.85 | 68.04 | 73.38 | 58.83 | 81.90 | 86.59 | 88.65 | 83.68 | 346.06 | 424.93 |
| **GPT-2** | | 50.86 | 64.19 | 70.30 | 56.41 | 79.21 | 85.31 | 85.88 | 81.24 | 346.75 | 423.22 |
| **w/o HTPP** | | 50.70 | 65.80 | 68.25 | 56.74 | 79.61 | 83.52 | 85.34 | 80.63 | 352.77 | 428.88 |
| **w/o VIMN** | | 51.12 | 65.07 | 71.47 | 57.46 | 80.67 | 85.01 | 86.52 | 81.24 | 348.51 | 428.38 |
| **w/o PPE** | | 50.61 | 64.44 | 71.83 | 57.49 | 80.42 | 86.10 | 86.68 | 82.83 | 348.52 | 424.52 |
| **w/o LLM** | | 45.21 | 56.68 | 64.05 | 51.10 | 74.23 | 81.78 | 81.99 | 77.63 | 358.04 | 438.41 |

### E.3.2 Rusults of Variants on BrightKite

**Language Model Variants.** We compared various language model variants on the Brightkite dataset across different tasks, as shown in Tab. 9. The TinyLlama (default) variant achieved the best performance in LP tasks, with the highest Acc@1 (**53.18%**) and Acc@20 (**74.11%**), and the second-best MRR (**59.89%**). It also excelled in TUL tasks, achieving the highest scores across all metrics, indicating its robust performance. LiteLlama performed exceptionally well, securing the highest Acc@5 (**69.13%**) and MRR (**59.98%**) in LP tasks, and the best MAE in TP tasks.

TinyLlama-Chat showed slightly lower performance compared to TinyLlama, particularly in TUL tasks, where it fell behind the other variants. Pythia-70M, despite having fewer parameters, performed competitively, achieving the second-best Acc@5 (**87.63%**) in TUL tasks and the best RMSE in TP tasks. Pythia-2.8B demonstrated strong performance as well, securing the second-best Acc@20 (**73.38%**) in LP tasks and showing consistency in TUL tasks.

Overall, the results suggest that the TinyLlama variants, especially the default one, are highly effective across different tasks, while LiteLlama and Pythia variants show specific strengths, indicating that the choice of backbone model can be tailored to the specific requirements of the task.

**Ablation Study.** The ablation study results in Tab. 9 highlight the importance of different components in the model. Removing HTPP (**w/o HTPP**) led to a significant drop in performance across all tasks. In TUL tasks, MRR decreased of **5.95%**, while in LP tasks, there was a notable decrease in performance metrics, with Acc@20 decreased of **7.91%** and MRR of **5.26%** indicating the critical role of HTPP in maintaining prediction accuracy.

Removing VIMN (**w/o VIMN**) also resulted in decreased performance, particularly in LP and TUL tasks, where the metrics showed a notable average performance degradation of **4.07%**.

The absence of PPE (**w/o PPE**) led to moderate performance degradation, with MRR in LP tasks decreased of **4.01%** and in TUL tasks of **3.38%**. The removal of LLM (**w/o LLM**) had the most pronounced negative effect, especially in LP tasks where Acc@5 decreased of **17.03%** and MRR of **14.68%**. This underscores the importance of the LLM component, as its absence significantly hinders the model's overall performance across all tasks.

Table 10: Ablations on FourSquare dataset in all tasks. **Red**: the best, Blue: the second-best.

| Tasks | | LP | | | | TUL | | | | TP | |
| --- | --- | --- | --- | --- | --- | --- | --- | --- | --- | --- | --- |
| Variant | Metic | Acc@1 | Acc@5 | Acc@20 | MRR | Acc@1 | Acc@5 | Acc@20 | MRR | MAE | RMSE |
| **TinyLlama (default)** | | **17.29** | 37.17 | **53.16** | 26.47 | 72.08 | **79.67** | 84.32 | **75.71** | 309.78 | 505.03 |
| **TinyLlama-Chat** | | 16.88 | 35.94 | 52.18 | 25.88 | 64.37 | 73.50 | 79.63 | 68.54 | 310.68 | **507.17** |
| **LiteLlama** | | 16.80 | **37.50** | 52.10 | **26.59** | **72.10** | 78.42 | 81.83 | 73.46 | 311.43 | 489.99 |
| **phi-2** | | 16.36 | 35.81 | 52.48 | 25.37 | 64.76 | 73.36 | 79.55 | 68.69 | **313.34** | 506.72 |
| **pythia-70M** | | 16.07 | 36.83 | 52.04 | 25.97 | 71.01 | 78.79 | 82.30 | 74.43 | 311.94 | 507.07 |
| **pythia-1B** | | 16.23 | 36.84 | 52.11 | 26.12 | 70.65 | 78.08 | **84.39** | 73.87 | 308.51 | 507.01 |
| **pythia-2.8B** | | 17.13 | 37.28 | 52.95 | 26.16 | 71.58 | 77.77 | 82.65 | 73.31 | 310.69 | 505.50 |
| **GPT-2** | | 16.47 | 34.96 | 50.38 | 24.91 | 69.22 | 76.55 | 80.39 | 71.89 | 310.67 | 506.50 |
| **w/o HTPP** | | 16.55 | 35.91 | 49.25 | 25.23 | 68.94 | 75.55 | 80.05 | 70.78 | 314.81 | 512.76 |
| **w/o VIMN** | | 16.62 | 35.51 | 51.06 | 25.27 | 69.79 | 76.21 | 80.26 | 71.60 | 311.63 | 510.11 |
| **w/o PPE** | | 16.51 | 35.14 | 51.73 | 25.36 | 70.00 | 77.42 | 80.81 | 72.86 | 310.70 | 506.53 |
| **w/o LLM** | | 14.65 | 30.78 | 45.95 | 22.58 | 64.28 | 73.46 | 76.52 | 68.62 | 321.44 | 519.45 |

### E.3.3 Rusults of Variants on Foursquare

**Language Model Variants.** We evaluated multiple language model variants on the FourSquare dataset across different tasks, with results presented in Tab. 10. The TinyLlama (default) variant demonstrated the best performance in LP tasks, achieving the highest Acc@1 (**17.29%**), Acc@20 (**53.16%**), and the second-best MRR. In TUL tasks, it also performed exceptionally well, securing the second-best Acc@1 (**72.08%**) and the best Acc@20 (**84.32%**) and MRR. This indicates its overall robustness in handling various tasks.

The LiteLlama variant excelled with the highest Acc@5 (**37.50%**) and MRR in LP tasks and matched TinyLlama in TUL tasks with the best Acc@1 (**72.10%**), although it slightly underperformed in other TUL metrics. TinyLlama-Chat, while showing strong performance, did not outperform the default TinyLlama variant, particularly in TUL tasks where it scored lower across all metrics.

Pythia-70M and Pythia-2.8B showed competitive results, with Pythia-70M achieving the second-best Acc@5 (**78.79%**) in TUL tasks and Pythia-2.8B securing the second-best Acc@1 (**17.13%**) in LP

tasks. Pythia-1B also performed well, particularly in TUL tasks with the highest Acc@20 (**84.39%**). GPT-2, while generally performing well, did not achieve top ranks in any specific metric, indicating it may not be the best choice for these tasks compared to other variants.

**Ablation Study.** The ablation study in Tab. 10 highlights the importance of various components in the model. Removing HTPP (**w/o HTPP**) led to a noticeable drop in performance across all tasks. For instance, in LP tasks, Acc@20 decreased of **7.35%** and MRR of **4.69%**, while in TUL tasks, MRR decreased of **6.51%**, underscoring the critical role of HTPP in maintaining high performance.

Removing VIMN (**w/o VIMN**) also resulted in decreased performance, particularly in LP and TUL tasks, where metrics showed a notable average performance degradation of **4.32%**.

The absence of PPE (**w/o PPE**) led to moderate performance degradation, with MRR in LP tasks decreased of **4.19%** and in TUL tasks of **3.76%**. The removal of LLM (**w/o LLM**) had the most significant negative effect, especially in LP tasks where Acc@5 decreased of **17.19%** and MRR of **14.69%**. This emphasizes the importance of the LLM component, as its absence significantly hinders the model's overall performance across all tasks.

# F  Few Shot Study

**The results of 1% few-shot learning** are presented in Tab. 11 and they significantly outperform all baseline methods. We attribute this to the efficient knowledge activation in our reprogrammed LLM. Remarkably, our approach consistently surpasses other competitive baselines across both LP and TUL tasks, further highlighting the potential of language models in proficient human behavior analysis. When compared to recent state-of-the-art models such as NSTPP, DSTPP, S2TUL, ReMVC, and CACSR, our average improvements exceed 20.3%, 22.9%, 84.5%, 48.2%, and 41.2% respectively, across all metrics. Even with only 1% of the training dataset, our model achieves results comparable to other models using 100% of the training dataset. This is particularly significant for privacy-protected and typically smaller Check-in datasets, as our model can effectively understand the distribution patterns of human behaviors with minimal data.

Table 11: Few-shot learning on 1% training data. Red: the best, Blue: the second-best.

| Task | Method | Gowalla Acc@1 | Acc@5 | Acc@20 | MRR | WeePlace Acc@1 | Acc@5 | Acc@20 | MRR | Brightkite Acc@1 | Acc@5 | Acc@20 | MRR | FourSquare Acc@1 | Acc@5 | Acc@20 | MRR |
|---|---|---|---|---|---|---|---|---|---|---|---|---|---|---|---|---|---|
| LP | DeepMove | 5.03 | 11.55 | 16.72 | 8.18 | 13.22 | 29.00 | 41.96 | 20.33 | 33.55 | 47.07 | 52.57 | 40.67 | 11.28 | 26.19 | 36.59 | 14.65 |
| | LightMove | 4.68 | 10.33 | 14.84 | 7.40 | 13.04 | 28.98 | 42.41 | 20.81 | 33.13 | 44.57 | 50.90 | 39.45 | 10.05 | 22.95 | 32.30 | 12.92 |
| | PLSPL | 5.49 | 11.98 | 16.75 | 8.57 | 12.38 | 27.64 | 41.58 | 20.03 | 34.91 | 46.42 | 52.71 | 41.05 | 12.77 | 28.48 | 38.46 | 16.32 |
| | HMT-LSTM | 5.12 | 11.08 | 16.20 | 8.07 | 11.93 | 27.03 | 40.57 | 19.53 | 33.08 | 45.02 | 50.54 | 39.65 | 10.31 | 23.58 | 33.07 | 13.32 |
| | LSTPM | 4.68 | 10.32 | 14.94 | 7.46 | 13.60 | 28.20 | 40.67 | 21.15 | 28.85 | 38.64 | 44.55 | 34.46 | 11.68 | 26.78 | 37.87 | 15.19 |
| | VaSCL | 5.35 | 11.02 | 16.31 | 8.20 | 12.49 | 29.04 | 42.13 | 21.30 | 33.87 | 46.76 | 52.71 | 40.96 | 11.36 | 25.79 | 36.43 | 14.66 |
| | SimCSE | 3.39 | 8.00 | 11.73 | 5.70 | 9.67 | 23.86 | 37.20 | 16.81 | 31.60 | 45.86 | 52.22 | 38.79 | 11.00 | 25.97 | 37.37 | 14.58 |
| | NSTPP | 5.12 | 11.49 | 16.30 | 8.30 | 11.46 | 25.04 | 38.13 | 18.30 | 31.04 | 41.40 | 47.58 | 37.21 | 12.33 | 27.70 | 37.79 | 15.74 |
| | DSTPP | 5.14 | 11.42 | 16.46 | 8.25 | 12.62 | 28.14 | 41.94 | 20.41 | 32.74 | 44.49 | 50.15 | 39.19 | 10.05 | 23.15 | 32.09 | 12.96 |
| | ReMVC | 5.25 | 11.32 | 16.44 | 8.22 | 12.96 | 28.74 | 42.72 | 20.96 | 33.58 | 44.95 | 51.41 | 40.18 | 10.44 | 24.24 | 33.52 | 13.42 |
| | SML | 4.70 | 10.33 | 14.90 | 7.52 | 11.92 | 26.24 | 39.65 | 19.36 | 31.21 | 41.79 | 48.54 | 36.93 | 11.12 | 25.69 | 36.39 | 14.68 |
| | CACSR | 5.21 | 9.09 | 13.10 | 6.34 | 10.70 | 23.88 | 36.66 | 17.55 | 30.13 | 44.01 | 48.96 | 36.96 | 11.13 | 24.87 | 36.29 | 14.26 |
| | **Mobility-LLM** | **6.59** | **13.95** | **20.37** | **10.26** | **15.63** | **32.39** | **47.67** | **23.64** | **38.62** | **50.71** | **58.60** | **44.61** | **13.16** | **29.52** | **42.38** | **16.78** |
| TUL | TULER | 18.54 | 29.77 | 36.15 | 22.95 | 34.62 | 52.05 | 65.59 | 43.70 | 33.31 | 47.59 | 63.44 | 40.41 | 19.29 | 29.78 | 41.20 | 26.89 |
| | TULVAE | 19.21 | 31.59 | 38.26 | 24.56 | 40.79 | 56.07 | 66.97 | 46.84 | 16.72 | 26.06 | 39.11 | 21.19 | 8.60 | 12.81 | 18.26 | 12.56 |
| | Movesim | 8.70 | 20.24 | 28.56 | 13.93 | 31.48 | 46.61 | 59.52 | 38.02 | 27.43 | 39.39 | 55.82 | 33.51 | 25.72 | 33.78 | 42.80 | 22.71 |
| | S2TUL | 22.66 | 34.06 | 36.98 | 27.61 | 27.95 | 36.42 | 43.22 | 30.79 | 17.29 | 25.03 | 31.48 | 25.57 | 23.75 | 29.42 | 38.06 | 32.53 |
| | VaSCL | 13.32 | 19.51 | 21.62 | 15.75 | 17.27 | 24.77 | 32.82 | 21.76 | 27.11 | 41.51 | 60.90 | 33.01 | 22.68 | 32.65 | 43.39 | 28.90 |
| | SimCSE | 14.98 | 26.27 | 34.50 | 19.61 | 29.43 | 47.19 | 62.34 | 38.01 | 27.77 | 41.18 | 57.68 | 34.02 | 17.05 | 26.60 | 39.51 | 22.94 |
| | ReMVC | 19.50 | 30.90 | 33.25 | 24.39 | 35.72 | 49.10 | 54.24 | 39.41 | 28.83 | 38.70 | 48.21 | 33.75 | 16.29 | 25.74 | 36.37 | 22.70 |
| | SML | 18.91 | 28.68 | 31.37 | 24.80 | 31.54 | 47.45 | 53.75 | 37.44 | 29.01 | 40.91 | 51.54 | 35.19 | 15.99 | 22.74 | 25.54 | 30.41 |
| | CACSR | 17.29 | 28.73 | 35.13 | 22.83 | 38.01 | 53.99 | 66.95 | 45.77 | 26.45 | 41.88 | 57.79 | 33.65 | 21.59 | 30.92 | 40.62 | 29.81 |
| | **Mobility-LLM** | **39.28** | **50.50** | **52.81** | **44.98** | **51.79** | **68.14** | **77.66** | **59.32** | **46.48** | **57.77** | **70.23** | **52.67** | **44.69** | **51.83** | **59.18** | **52.94** |

**The results of 20% few-shot learning** are presented in Tab. 12 and they significantly outperform all baseline methods. We attribute this to the efficient knowledge activation in our reprogrammed LLM. Remarkably, our approach consistently surpasses other competitive baselines across both LP and TUL tasks, further highlighting the potential of language models in proficient human behavior analysis. When compared to recent state-of-the-art models such as NSTPP, DSTPP, S2TUL, ReMVC, and CACSR, our average improvements exceed 21.4%, 21.8%, 86.6%, 46.3%, and 45.1% respectively, across all metrics.

Table 12: Few-shot learning on 20% training data. Red: the best, Blue: the second-best.

| Task | Method | Gowalla Acc@1 | Acc@5 | Acc@20 | MRR | WeePlace Acc@1 | Acc@5 | Acc@20 | MRR | Brightkite Acc@1 | Acc@5 | Acc@20 | MRR | FourSquare Acc@1 | Acc@5 | Acc@20 | MRR |
|---|---|---|---|---|---|---|---|---|---|---|---|---|---|---|---|---|---|
| LP | DeepMove | 8.19 | 19.09 | 27.59 | 13.10 | 16.66 | 33.80 | 47.76 | 24.33 | 44.48 | 61.15 | 64.51 | 52.28 | 14.67 | 32.69 | 44.12 | 23.06 |
| | LightMove | 7.67 | 17.11 | 24.30 | 11.89 | 16.24 | 33.71 | 48.66 | 24.63 | 43.79 | 58.49 | 62.28 | 51.07 | 13.12 | 28.62 | 39.15 | 20.31 |
| | PLSPL | 8.98 | 19.82 | 27.41 | 13.82 | 15.65 | 31.94 | 47.42 | 23.85 | 46.10 | 60.25 | 64.50 | 53.19 | 16.67 | 35.66 | 46.37 | 25.64 |
| | HMT-LSTM | 8.41 | 18.28 | 26.69 | 12.97 | 15.13 | 31.20 | 46.50 | 23.41 | 44.17 | 58.84 | 61.28 | 50.92 | 13.51 | 29.52 | 40.08 | 20.78 |
| | LSTPM | 7.70 | 17.15 | 24.50 | 11.96 | 16.97 | 32.58 | 46.85 | 25.16 | 38.33 | 50.60 | 54.51 | 44.66 | 15.21 | 33.63 | 45.85 | 23.86 |
| | VaSCL | 8.75 | 18.19 | 26.68 | 13.11 | 15.66 | 33.48 | 48.39 | 25.64 | 44.96 | 61.42 | 63.85 | 52.70 | 14.80 | 32.42 | 44.33 | 22.93 |
| | SimCSE | 5.61 | 13.25 | 19.37 | 9.14 | 12.24 | 27.53 | 42.56 | 20.10 | 41.61 | 59.46 | 63.76 | 50.31 | 14.28 | 32.64 | 45.15 | 23.01 |
| | NSTPP | 8.45 | 18.96 | 26.73 | 13.29 | 14.31 | 28.87 | 43.36 | 21.72 | 41.00 | 53.83 | 58.39 | 48.37 | 16.06 | 34.65 | 45.62 | 24.62 |
| | DSTPP | 8.42 | 18.85 | 27.01 | 13.17 | 15.86 | 32.51 | 48.22 | 24.16 | 43.50 | 57.81 | 61.05 | 50.99 | 13.05 | 28.73 | 39.04 | 20.48 |
| | ReMVC | 8.55 | 18.77 | 27.06 | 13.10 | 16.13 | 33.04 | 48.54 | 25.10 | 44.44 | 58.57 | 62.65 | 52.17 | 13.45 | 30.02 | 40.59 | 21.16 |
| | SML | 7.70 | 17.12 | 24.75 | 11.98 | 15.01 | 30.47 | 45.08 | 23.05 | 41.43 | 54.45 | 59.21 | 47.76 | 14.49 | 32.09 | 44.15 | 23.02 |
| | CACSR | 8.50 | 14.96 | 21.62 | 10.13 | 13.49 | 27.67 | 41.61 | 21.02 | 39.83 | 57.58 | 59.73 | 48.04 | 14.54 | 31.08 | 43.63 | 22.33 |
| | **Mobility-LLM** | 10.79 | 23.01 | 33.50 | 16.44 | 19.62 | 37.42 | 54.37 | 28.20 | 51.26 | 66.08 | 71.41 | 57.75 | 17.09 | 36.92 | 51.36 | 26.28 |
| TUL | TULER | 30.98 | 44.60 | 55.36 | 35.70 | 45.96 | 60.56 | 71.54 | 54.71 | 49.63 | 65.99 | 76.73 | 57.09 | 23.33 | 37.81 | 51.72 | 30.56 |
| | TULVAE | 31.87 | 47.57 | 58.55 | 38.40 | 54.49 | 65.23 | 73.49 | 58.51 | 25.11 | 36.06 | 47.15 | 29.99 | 10.46 | 16.48 | 22.71 | 14.39 |
| | Movesim | 14.41 | 30.36 | 43.83 | 21.69 | 41.71 | 54.61 | 64.79 | 47.60 | 41.35 | 54.17 | 67.45 | 47.62 | 31.30 | 43.70 | 53.66 | 25.89 |
| | S2TUL | 37.93 | 51.02 | 56.59 | 42.83 | 37.18 | 42.55 | 47.10 | 38.32 | 26.14 | 34.50 | 37.84 | 36.13 | 29.05 | 37.54 | 47.77 | 37.12 |
| | VaSCL | 22.30 | 29.33 | 33.19 | 24.57 | 22.95 | 28.91 | 36.22 | 27.13 | 40.84 | 57.56 | 73.50 | 46.73 | 27.58 | 41.90 | 54.84 | 32.72 |
| | SimCSE | 24.98 | 39.28 | 52.73 | 30.65 | 38.99 | 55.56 | 68.14 | 47.62 | 41.83 | 56.92 | 69.55 | 47.91 | 20.77 | 33.93 | 49.59 | 26.03 |
| | ReMVC | 32.74 | 46.30 | 51.24 | 37.93 | 47.56 | 57.12 | 59.40 | 49.33 | 43.78 | 53.12 | 58.14 | 47.68 | 19.73 | 32.71 | 45.78 | 25.96 |
| | SML | 31.61 | 42.97 | 47.96 | 38.35 | 41.83 | 55.48 | 58.62 | 46.86 | 43.83 | 56.67 | 62.09 | 49.86 | 19.32 | 29.11 | 32.13 | 34.70 |
| | CACSR | 28.98 | 43.44 | 53.54 | 35.62 | 50.37 | 62.81 | 73.60 | 57.41 | 39.92 | 57.77 | 69.33 | 47.83 | 26.28 | 39.81 | 51.34 | 33.88 |
| | **Mobility-LLM** | 65.55 | 75.74 | 80.97 | 70.25 | 68.90 | 79.75 | 84.96 | 74.03 | 69.94 | 79.86 | 84.60 | 74.56 | 54.39 | 66.46 | 74.28 | 60.17 |

# G  Feed Forward Layer

We aim to effectively fuse the sequences $[z_{i-r+1}, \cdots, z_{i-1}, z_i]$ and $[s_{i-r+1}, \cdots, s_{i-1}, s_i]$ using a gated feed-forward layer inspired by transformers. The fusion mechanism is designed to be sophisticated while maintaining the core principles of the feed-forward structure.

**Gating Mechanism and Gated Inputs:** To enhance the fusion process, we introduce gating mechanisms $\mathbf{g}_x$ and $\mathbf{g}_z$ to control the flow of information from input vectors $\mathbf{x}$ and $\mathbf{z}$. Then, apply the gating mechanisms to the inputs:

$$
\begin{aligned}
\mathbf{g}_x = \sigma(\mathbf{W}_{g_x}\mathbf{x} + \mathbf{b}_{g_x}), \mathbf{g}_z = \sigma(\mathbf{W}_{g_z}\mathbf{z} + \mathbf{b}_{g_z}), \\
\mathbf{x}_g = \mathbf{g}_x \odot \mathbf{x}, \mathbf{z}_g = \mathbf{g}_z \odot \mathbf{z},
\end{aligned}
\tag{8}
$$

where $\sigma$ is the sigmoid activation function, $\mathbf{W}_{g_x} \in \mathbb{R}^{d_g \times d_x}$, $\mathbf{b}_{g_x} \in \mathbb{R}^{d_g}$, $\mathbf{W}_{g_z} \in \mathbb{R}^{d_g \times d_z}$, and $\mathbf{b}_{g_z} \in \mathbb{R}^{d_g}$, and $\odot$ denotes element-wise multiplication.

**Multi-Layer Perceptron (MLP), Layer Normalization, and Residual Connection:**

Concatenate the gated inputs and process them through a multi-layer perceptron, apply layer normalization to stabilize the training process, and incorporate a residual connection to preserve the original information:

$$
\begin{aligned}
\mathbf{h}_1 = \text{ReLU}(\mathbf{W}_1[\mathbf{x}_g; \mathbf{z}_g] + \mathbf{b}_1), \mathbf{h}_2 = \text{ReLU}(\mathbf{W}_2\mathbf{h}_1 + \mathbf{b}_2), \\
\mathbf{h}_{\text{norm}} = \text{LayerNorm}(\mathbf{h}_2), \mathbf{h}_{\text{res}} = \mathbf{h}_{\text{norm}} + [\mathbf{x}_g; \mathbf{z}_g],
\end{aligned}
\tag{9}
$$

where $\mathbf{W}_1 \in \mathbb{R}^{d_1 \times (d_x + d_z)}$, $\mathbf{b}_1 \in \mathbb{R}^{d_1}$, $\mathbf{W}_2 \in \mathbb{R}^{d_2 \times d_1}$, $\mathbf{b}_2 \in \mathbb{R}^{d_2}$.

Perform a final linear transformation to produce the fused output:

$$
\boldsymbol{h}_i = \mathbf{W}_3\mathbf{h}_{\text{res}} + \mathbf{b}_3,
\tag{10}
$$

where $\mathbf{W}_3 \in \mathbb{R}^{d_f \times (d_x + d_z)}$, $\mathbf{b}_3 \in \mathbb{R}^{d_f}$.

This design leverages the powerful feed-forward layer structure with enhancements like gating mechanisms, MLPs, layer normalization, and residual connections to achieve a sophisticated and effective fusion of the input sequences $[z_{i-r+1}, \cdots, z_{i-1}, z_i]$ and $[s_{i-r+1}, \cdots, s_{i-1}, s_i]$.

# H GeoHash Embedding Layer

The Geohash embedding function converts geographic coordinates (latitude and longitude) into a Geohash-encoded embedding vector. This involves two primary steps: first, encoding the latitude and longitude into a Geohash string, and second, converting this Geohash string into an embedding vector.

**Geohash Encoding** Given a latitude $L_{lat}$ and longitude $L_{lon}$, the Geohash encoding function $G$ maps the coordinates to a Geohash string $g$:

$$g = G(L_{lat}, L_{lon}) \tag{11}$$

The function $G$ performs the following operations: Interleave the binary representations of latitude and longitude. 2. Encode the resulting binary string into base32 characters to form the Geohash string $g$.

**Geohash to Embedding Vector** Let $\mathcal{E}$ be the embedding function that converts a Geohash string $g$ into an embedding vector $\mathbf{v}$ in $\mathbb{R}^d$:

$$\mathbf{v} = \mathcal{E}(g) \tag{12}$$

The embedding function $\mathcal{E}$ can be implemented using various methods such as a lookup table or a neural network that maps the discrete Geohash values to continuous vector spaces.

**Combined Function** Combining the two steps, we define the overall Geohash embedding function $\mathcal{F}$:

$$\mathcal{F}(L_{lat}, L_{lon}) = \mathcal{E}(G(L_{lat}, L_{lon})) \tag{13}$$

This can be expanded as:

$$\mathbf{v} = \mathcal{F}(L_{lat}, L_{lon}), \tag{14}$$

where $\mathcal{F}$ first encodes the latitude $L_{lat}$ and longitude $L_{lon}$ into a Geohash string using $G$, and then maps the Geohash string to an embedding vector using $\mathcal{E}$.

# I List of POI Categories

Airport, Bank, Bakery, Beach, Bridge, Cafe, Cinema, Clinic, College, Church, Courthouse, Embassy, Firestation, Gym, Harbor, Hospital, Hotel, Library, Market, Mall, Museum, Office, Park, Pharmacy, Pub, Restaurant, School, Stadium, Station, Subway, Supermarket, Theater, University, Zoo, Alley, Aquarium, Arch, Art, Bakery, Bar, Basin, Bay, Bench, Bicycle, Boat, Border, Bowling, Brewery, Buffet, Bungalow, Butcher, Cabaret, Cabin, Canal, Candy, Casino, Castle, Cemetery, Circus, Cliff, Club, Coffeehouse, College, Court, Creek, Cruise, Dam, Dance, Deck, Diner, Dive, Dock, Dorm, Drive, Embassy, Factory, Farm, Fastfood, Ferry, Field, Fishing, Fitness, Fountain, Gallery, Garage, Garden, Gate, Gazebo, Grill, Guesthouse, Hike, Hostel, Ice, Inn, Island, Jail, Kiosk, Lake, Lane, Library, Lighthouse, Mansion, Marina, Meadow, Motel, Monument, Mountain, Nursery, Observatory, Opera, Orchard, Outpost, Palace, Pantry, Pier, Planetarium, Plaza, Pool, Post, Promenade, Pub, Ranch, Recreation, Refuge, Resort, Restaurant, Retreat, Roadhouse, Ruin, RV, Salon, Sanctuary, Sauna, Shelter, Shrine, Silo, Ski, Snack, Spa, Speedway, Spring, Square, Statue, Studio, Subway, Swim, Tavern, Temple, Terminal, Track, Trail, Tram, Tunnel, Vineyard, Warehouse, Wharf, Wildlife, Windmill, Winery, Yard, Yoga, Zoo, Abattoir, Aircraft, Amphitheater, Apartment, Arena, Auction, Auditorium, Avenue, Baggage, Barbecue, Bazaar, Boathouse, Buffet, Cafeteria, Campsite, Carpark, Carwash, Carousel, Chapel, Chemistry, Circus, Clinic, Clubhouse, Compound, Conservatory, Convent, Corner, Crematorium, Croft, Deli, Den, Dockyard, Driveway, Enclosure, Estate, Facility, Farmhouse, Festival, Fieldhouse, Firehouse, Florist, Forge, Fountain, Foundry, Gallery, Gasworks, Grotto, Gymnasium, Hall, Hangar, Harbormaster, Heritage, Homestead, Hospice, Hostel, House, Hut, Inn, Jamboree, Jetty, Junction, Kiosk, Laboratory, Lagoon, Lavatory, Library, Lighthouse, Lodge, Lookout, Mill, Mission, Motel, Office, Oratory, Pavilion, Plaza, Platform, Postbox, Range, Refinery, Reserve, Retreat,

Schoolhouse, Shop, Skatepark, Slope, Stand, Station, Synagogue, Tavern, Teahouse, Terrace, Tower, Treasury, Villa, Waterslide, Wharf, Workshop.

