# OpenReview forum: "Mobility-LLM: Learning Visiting Intentions and Travel Preference from Human Mobility Data with Large Language Models"
_NeurIPS.cc/2024/Conference — NeurIPS 2024 poster_

### Official Review · Reviewer_6rUV · 2024-06-13

**Soundness:** 3
**Presentation:** 3
**Contribution:** 2
**Rating:** 5
**Confidence:** 4

**Summary:**

This submission presents a framework that leverages a large language model (LLM) as the backbone architecture for human-mobility-related tasks, including user identification, next location prediction, and arrival time prediction. The authors introduce a POI (Point of Interest) Point-wise Embedding Layer and a Visiting Intention Memory Network as an encoder to process check-in data. Additionally, they establish a behavior prompt pool to incorporate behavior preference information with the embeddings from the Visiting Intention Memory Network. The framework is evaluated using four real-world datasets, and the proposed model outperforms baseline methods in terms of most metrics.

**Strengths:**

1.	Overall, this paper is well-written and well organized. The authors illustrate the motivation of this study sufficiently and review the related work thoroughly.
2.	The proposed framework finetunes the LLM using semantic data, which is novel in human mobility related tasks.
3.	The experimental studies are thorough and the hyperparameter analysis is well presented.

**Weaknesses:**

1.	While the framework offers a novel approach to addressing human mobility-related tasks, the robustness of the modules is questionable, particularly in the formulation of the human travel preference prompt. Additionally, the overall technical contribution is limited.
2.	The paper provides no discussion of choice of scoring function.
3.	There is a lack of novelty in PPEL and HTPP.

**Questions:**

1.	The reviewer suggests that the HTPP is the most significant idea in this study. However, However, several concerns need to be addressed: (i) The use of cos-similarity based scoring function to determine the significance score needs more illustration. Why is cos-similarity sufficient to evaluate the semantic similarity in urban mobility tasks? What is the impact of choice of the scoring function? (ii) What is the motivation of the selected domains as well as the prompt words? Are there any theoretical support?
2.	This paper essentially treats each domain separately during top-k prompts selection. The reviewer suggests that it is the combination of prompts of each domain that presents meaningful and promising semantic information, i.e., a doctor (Domain 2) goes to exercise (Domain 1) to keep healthy (Domain 3).

**Limitations:**

The authors have discussed the limitations adequately.

---

> ### Author Rebuttal · Authors · 2024-08-06
>
> Thank you for your thoughtful and constructive comments.
>
> ### **[W1&W2&Q1]**
>
> We design the VIMN module to reprogram the check-in sequences. After passing through the VIMN module, the check-in sequences are aligned to the natural language semantic space, producing the resulting $\mathbf{h}_i$*.* This ensures that the two representation vectors sent to the HTPP module are in the same semantic space. We believe that in the same semantic space, vectors that are close in distance have similar semantics. Therefore, if the cosine similarity calculated by the score-function is high, it indicates that $\mathbf{h}_i$ itself has semantic relevance in the semantic representation space. In summary, the key to evaluating the semantic similarity in urban mobility tasks lies in the design of the VIMN reprogramming module, with cos-similarity only serving to compare the similarity of vectors in the same semantic space.
>
> ### **[W2]**
>
> While the PPEL and HTPP modules leverage existing technologies from a technical perspective, our design principles are innovative and introduced for the first time in tasks related to human activities. These modules are integral, as they employ LLM's word token embeddings for both check-in sequence embeddings and prompt tokens, achieving semantic alignment and integration with other modules. Ablation studies have shown that these modules significantly enhance performance, thereby validating the efficacy and rationality of our design.
>
> ### **[Q2]**
>
> We considered the scheme you originally mentioned. However, listing all combinations creates a very large space, reducing the model's efficiency. Manually screening reasonable combinations will cost a huge amount of human resources and may fail to cover all possibilities due to the diversity of human activity data and the many situations that can arise. Additionally, case studies show that the existing scheme is superior in efficiency and accuracy compared to the one you mentioned.

---

> ### Comment · Reviewer_6rUV · 2024-08-08
>
> I am disappointed with the author's rebuttal. The author did not sufficiently address my questions; the responses lacked clarity and evaded the main issues. For instance, I don't understand the purpose of the combined response to W1&W2Q1. Your answer is completely off-topic, and I feel that W2's question should correspond to W3. Specifically, my questions are:
>
> 1. I have concerns about the robustness of the human travel preference prompt (HTPP) module. For example, your HTPP module includes several critical parameters, such as D. Why is the domain set to 3? Is this a limitation of the dataset, an arbitrary definition, or derived from experiments? I am unclear. What is the reason for setting m to 16? Even if the author did not mention this in the paper, it should be adequately addressed in the rebuttal. Unfortunately, I did not find any relevant answers.
>
> 2. My question is why you chose cosine similarity instead of other methods like Euclidean distance or Manhattan distance. These could all be discussed. Of course, I know that cosine similarity can measure the similarity of vectors in a unified semantic space, but what are its specific advantages? Did you determine through experiments that cosine similarity is better? Can't these topics be further discussed?
>
> 3. Your response claims that the existing solution is superior to the idea I proposed in Q2. The original text states, "We provide three case studies to visualize the improvements brought by HTPP and VIMN." I do not understand the relevance of the cases you provided to my question. I need a clearer explanation, such as comparing my question's definition with your case studies. Your response only added to my confusion.
>
> Other reviewers have also provided insights from different perspectives, for example:
>
> 1. Reviewer NycQ believes that the performance improvement is not significant enough, which is indeed a problem. Your rebuttal mentioned that the t-value between Mobility-LLM and the best baseline is very high. However, I think this is because both models are very stable, and changing the random seed does not overly affect the model's performance, leading to a low standard deviation and a high t-value. This is meaningless. The core issue is that your improvement over the best baseline is only 0.4%. While such an improvement might be acceptable for other deep learning papers, this paper is based on LLM, requiring better GPU, more computing time, and potential API costs, making the slight performance improvement questionable.
>
> 2. For example, the classic baseline Graph-Flashback [1] for next location prediction shows more than double the performance improvement on Gowalla compared to DeepMove, and 4% higher on Foursquare. Why didn't you compare with some classic baselines, or if your performance improvement could also reach this level, it would be more reasonable. DeepMove is only a baseline published in 2018, but according to Table 1, your model only shows about a 1% improvement over DeepMove on Gowalla and Foursquare.
>
> 3. The authors claim that their convergence speed is faster than other baselines, but I did not see any relevant evidence. If you are referring to fewer epochs, I think this is unreasonable. The time each model consumes per epoch is completely inconsistent. The speed of convergence should consider the overall training time. What I observed is that Mobility-LLM consumes more memory, requires more training time, and inference time. The increased computational cost only brings minimal performance gains.
>
> 4. Reviewer Mti9 considers your Figure 1 to be rather cluttered. Why didn't the authors use the global response to upload a PDF showing the improved figure? After your adjustments, we are not clear about what the final version will look like.
>
> 5. Additionally, CACSR was not discussed. Why didn't you present a relevant discussion in the rebuttal?
>
> Overall, although the authors initially received an overall positive comment, I do not see the authors taking the reviewers' questions seriously in the rebuttal, and the evasive answers have deepened my doubts. It is undeniable that the completion of this paper, including the production of charts and experimental results, is very high. However, the vague explanations of many details, the additional computational costs, and the insufficiently superior performance lead me to lean towards rejecting this paper, despite the positive attitude of other reviewers.
>
> [1] Rao, Xuan, et al. "Graph-flashback network for next location recommendation." Proceedings of the 28th ACM SIGKDD conference on knowledge discovery and data mining. 2022.

---

> > ### Author Response · Authors · 2024-08-10
> >
> > We sincerely apologize for not accurately addressing your question. We realize that we did not fully understand your comments before. In this discussion, we will first address the questions raised in your current comment in detail and then respond to the issues raised by other reviewers in the subsequent comments.
> >
> >
> > ### 【Yours-Q1】
> > The selection of domains in the HTTP module is based on our research summary of various materials and reports describing human activity characteristics. Initially, besides these three domains, we also considered other fields such as age, travel time, and activity range.
> >
> > After testing on a small dataset, we discovered that the selected three domains had the most significant impact on the model's performance. For each set of terms within one domain, we asked an LLM to provide a comprehensive set of commonly used terms for that domain.
> >
> > We found that selecting 16 terms sufficiently met our experimental needs. Choosing more could lead to redundancy, while fewer terms would fail to cover the domain knowledge comprehensively. Ultimately, this exploration led to the current experimental setup.
> >
> > ### 【Yours-Q2】
> > Cosine similarity measures the angle between two vectors rather than the distance, making it suitable for measuring directional similarity rather than magnitude. In high-dimensional spaces, text or user behavior data is represented as high-dimensional sparse vectors, making it more effective in capturing textual feature similarity without being disrupted by data sparsity. The direction of the vectors often reflects the semantic similarity of words or sentences.
> >
> > In contrast, Euclidean distance and Manhattan distance reflect absolute positional differences, which can be problematic in high-dimensional sparse data. The presence of zero dimensions might reduce similarity, leading to indistinguishable distance calculations that fail to capture actual similarity[1]. Therefore, it is more reasonable to use cosine similarity in the HTPP module compared to other measurement methods.
> >
> > [1] Turney, P. D., & Pantel, P. (2010). From frequency to meaning: Vector space models of semantics. Journal of Artificial Intelligence Research, 37, 141-188.
> >
> > ### 【Yours-Q3】
> > We acknowledge that our previous explanation may have been unclear, and we apologize for the confusion.
> >
> > **The case study mentioned in our previous response actually refers to the discussion in Q2 of the initial Rebuttal** and is not related to the case study in the paper.
> >
> > We would like to elaborate on the point we intended to express in the rebuttal as follows.
> >
> > At the outset of our model design, we tried three different approaches, including the one you mentioned in Q2:
> >
> > 1. Listing all possible combinations of all domains，each combination is composed of one term from each domain.
> > 2. Manually selecting reasonable combinations in approach 1.
> > 3. Our current approach.
> >
> > We compared the performance and efficiency of these three approaches. During the performance comparison, we looked into the results of approaches 1 and 3 through a case discussion. We found that the results from approach 3 were more reasonable, while the combinations from approach 1 were often unsatisfactory. During the training, the semantic of a sequence might match multiple terms in each domain, which means that if approach 1 was adopted, more than one combinations might be matched. Since the combinations contain semantic from multiple domains is coupled together in approach 1, this could cause interference among the matched combinations, resulting in unsatisfactory selections. However, in approach 3, each domain is decoupled and matched separately, so this issue does not exist.
> > Additionally, approach 1 was time-consuming, so after the case discussion, we discarded it.
> >
> > If the approach 2 is adopted, where multiple terms are selected from each domain, the number of combinations will be extremely large, making it impractical to manually select reasonable combinations.
> >
> > Therefore, approach 3 is optimal in terms of efficiency and performance, so we ultimately chose approach 3. This choice of approach and the above case discussion were not mentioned in the paper before. We will also add the above content to Section 5.5 of the paper. We are grateful for your suggestion, which helped us make our experiment and paper clearer and more complete.

---

> > ### Author Response · Authors · 2024-08-10
> >
> > ### 【Other-Q1】
> > We will explain from three aspects: parameter search, performance bottleneck, and multi-task framework.
> >
> > 1. To ensure fairness, we did not use default parameters for the experiments to save time. Instead, we conducted a comprehensive parameter search for all baseline models, selecting the optimal hyperparameters for each on different datasets to ensure they perform at their best. During the search, we found that some baseline models performed significantly better with certain unexpected parameter combinations (e.g., the DeepMove model's performance improved by 37% when the Dropout parameter was set to 0.9 compared to the default parameters). Therefore, the results we reported reflect the best possible performance of the baseline models, surpassing their default settings.
> > 2. To verify the effectiveness of our model on a larger dataset, we increased the sample size by including users with fewer than 10 check-ins (while other papers typically exclude users with fewer than 50 or 100 check-ins). This inclusion of more sparse samples makes further prediction improvements more difficult. The performance gap between these baselines is minimal, reaching a bottleneck where further improvements with conventional deep learning models are challenging. Thus, the current performance improvements are considered valuable.
> > 3. Our model can handle multi-task frameworks. To demonstrate its powerful performance, we compared it with multi-task framework models(such as ReMVC, VaSCL, SML, CACSR) and various SOTA models designed for specific downstream tasks. Our model achieved SOTA results in nearly all metrics, with improvements exceeding 26% in some cases, and outperformed other SOTA multi-task framework models by over 15%.

---

> > ### Author Response · Authors · 2024-08-10
> >
> > ### 【Other-Q2】
> > #### 【Other-Q2-1】
> > We first explain why we did not include Graph-Flashback as a baseline model from the perspectives of application scenarios, data processing methods, model architecture input, and dataset support.
> >
> > 1. **Application Scenario**: Graph-Flashback is a classic and effective baseline model, but it requires a fixed-length historical sequence as input. Our study focuses on the representation and prediction of variable-length sequences. Therefore, the baselines in our paper are models that can handle variable-length sequences.
> > 2. **Data Processing Methods**: In next location prediction research, it is common to handle variable-length sequences. Processing samples often involves filtering out POIs with less than 10 visits, users with less than 10 check-ins, limiting historical days to no more than 30, and ensuring that the next location visit interval does not exceed one day to qualify as a label. This approach leads to variable-length sample sequences that better reflect real-world scenarios. In contrast, the data processing method of Graph-Flashback, as mentioned in the original paper, is: "We discard inactive users who have less than 100 check-ins and sort each user’s check-ins in ascending order of timestamp. The first 80% check-ins of each user are split into multiple length-equally (e.g., 20) sequences, which are chosen as the training set. Likewise, the remaining 20% are regarded as the testing set."
> > 3. **Model Architecture Input**: The input to this model is a fixed-length sequence. We thoroughly reviewed the code of Graph-Flashback, and the historical sequences it processes are fixed-length sequences of 20. We also attempted to modify it into a variable-length prediction model, but since the entire framework is designed for fixed-length sequences, almost the entire framework would need adjustment, and many components can only handle fixed-length sequences. Further modifications would cause the model to deviate significantly from the original design of Graph-Flashback.
> > 4. **Dataset Support**: Not all datasets provide friendship information. Only the Gowalla and Foursquare datasets provide friendship networks, while Weeplace and BrightKite do not disclose friendship networks due to privacy reasons, or the disclosed friendship networks are unreliable. Therefore, Graph-Flashback cannot run on our other two datasets.
> >
> > We firmly believe that Graph-Flashback is a classic baseline model, and we have cited it in the Related Work section and will include Graph-flashback in our related work and make fully discussion in our revised version. However, due to the multiple issues mentioned above that cannot be unified or fairly addressed, Graph-Flashback could not be fairly included in our baseline model comparison.
> >
> > #### 【Other-Q2-2】
> > Next, we explain why the comparison between our model, Graph-Flashback, and DeepMove is not valid:
> >
> > 1. **Different Scenario Settings**: The models we selected, such as DeepMove, are designed to handle variable-length sequences, and their advantages are more evident in such scenarios. Although Graph-Flashback includes many baseline models that handle variable-length sequences, these baselines, including DeepMove, cannot fully exploit their advantages in the scenario set by Graph-Flashback. In the comparison experiment under Graph-Flashback's setting, the gaps between the baselines are large, while in our setting, the gaps are small. These differences are not on the same scale, so a direct comparison cannot distinguish the superiority of the models.
> > 2. **Different Experimental Settings**: We were surprised to find that the DeepMove model's performance improved by 37% when the Dropout parameter was set to 0.9. Although this parameter may seem unreasonable, we still respect the facts. However, we are unsure whether Graph-Flashback conducted experimental adjustments and a comprehensive parameter search for the Dropout parameter of DeepMove during the experiment.
> > 3. **Unified Framework**: Our model is a unified framework capable of handling multiple tasks. To demonstrate the powerful performance of our framework, we compared it with the most classic and SOTA end-to-end models in various downstream tasks. Therefore, more attention should be paid to the overall performance improvement.

---

> > ### Author Response · Authors · 2024-08-10
> >
> > ### 【Other-Q3】
> > As you mentioned, the number of epochs alone cannot effectively demonstrate the convergence speed. The total training time should be used to prove the speed of convergence. In further experiments, we tried reducing the early stopping patience from 10 to 3 to decrease the model's training time. So, we adjusted the patience to 3 and were surprised to find that some of the model's metrics actually improved with less traing time. After our analysis, we found that the reason is that the patience setting during the training of our model was relatively large compared to the total number of training epochs. This led to some unnecessary time overhead and caused our model to slightly overfit.
> >
> > Below, we present the GPU memory usage, training time for the LP task on the WeePlace dataset, and the results of the LP task under the new setting on four datasets. We also included variants of Python-70M (as presented in Table 5 and A.6 of the paper).
> >
> > |Method|Memory|Training Time|Training Epoch|
> > |-|-|-|-|
> > |PLSPL-Patience 10|3.4G|4.73h|79|
> > |PLSPL-Patience 3|3.4G|4.45h|75|
> > |Mobility-LLM (TinyLlama)-Patience 10|11.3G|5.45h|17|
> > |Mobility-LLM (TinyLlama)-Patience 3|11.3G|2.96h|10|
> > |Mobility-LLM (pythia-70M)-Patience 10|2.74G|2.76h|39|
> > |Mobility-LLM (pythia-70M)-Patience 3|2.74G|1.47h|21|
> >
> > |Datasets|Gowalla||||WeePlace||||Brightkite||||FourSquare||||
> > |-|-|-|-|-|-|-|-|-|-|-|-|-|-|-|-|-|
> > |Metric (e-2)|Acc@1|Acc@5|Acc@20|MRR|Acc@1|Acc@5|Acc@20|MRR|Acc@1|Acc@5|Acc@20|MRR|Acc@1|Acc@5|Acc@20|MRR|
> > |The Best Baseline|11.47±0.03|24.12±0.17|33.82±0.15|17.44±0.05|19.66±0.04|37.68±0.18|53.44±0.13|28.46±0.06|51.42±0.12|66.25±0.08|71.46±0.11|57.59±0.14|16.92±0.04|36.05±0.04|49.39±0.16|26.02±0.04|
> > |Mobility-LLM (TinyLlama)-Patience 10|11.87±0.04|25.14±0.04|36.36±0.09|18.29±0.11|20.47±0.13|39.22±0.19|56.69±0.10|29.21±0.15|53.18±0.17|68.31±0.14|74.11±0.18|59.89±0.03|17.29±0.03|37.17±0.02|53.16±0.20|26.47±0.05|
> > |Mobility-LLM (TinyLlama)-Patience 3|11.90±0.03|25.17±0.05|36.43±0.11|18.30±0.10|20.50±0.07|39.27±0.17|56.71±0.09|29.24±0.13|53.18±0.16|68.34±0.11|74.17±0.18|59.91±0.04|17.31±0.04|37.20±0.03|53.19±0.22|26.44±0.02|
> > |Mobility-LLM (pythia-70M)-Patience 10|11.03±0.16|24.86±0.11|35.74±0.07|17.91±0.17|19.88±0.18|38.02±0.03|54.19±0.04|28.46±0.13|52.03±0.10|66.34±0.02|72.09±0.05|57.93±0.22|17.01±0.04|36.20±0.09|50.88±0.13|25.84±0.16|
> > |Mobility-LLM (pythia-70M)-Patience 3|11.29±0.13|24.98±0.18|35.56±0.16|17.98±0.02|19.93±0.11|38.21±0.22|54.69±0.03|28.79±0.07|52.14±0.04|66.87±0.10|73.21±0.17|58.90±0.05|17.12±0.09|36.33±0.04|52.13±0.04|26.03±0.14|
> >
> > The experimental results show that our model can maintain its previous advantages while using less training time than the conventional deep learning models, with some metrics even improving.
> >
> > The new experimental setup shows that the TinyLlama base model can achieve better results in less time. We also conducted further experiments with a variant of pythia-70M and found that setting patience to 10 indeed caused slight overfitting. Our pythia-70M base model can outperform the PLSPL baseline model with less time and memory usage. To verify whether PLSPL also suffers from overfitting, we conducted experiments with patience set to 3, and the results are as follows:
> >
> > |Datasets|Gowalla||||WeePlace||||Brightkite||||FourSquare||||
> > |-|-|-|-|-|-|-|-|-|-|-|-|-|-|-|-|-|
> > |PLSPL-Patience 10|11.26±0.02|24.12±0.17|33.82±0.15|17.44±0.05|18.77±0.11|37.31±0.02|53.31±0.18|27.86±0.10|51.42±0.12|65.34±0.09|71.46±0.11|57.59±0.14|13.73±0.07|30.65±0.10|43.18±0.09|21.42±0.04|
> > |PLSPL-Patience 3|11.22±0.07|24.08±0.17|33.81±0.02|17.40±0.02|18.81±0.14|37.26±0.10|53.31±0.09|27.84±0.10|51.46±0.09|65.31±0.04|71.44±0.11|57.57±0.18|13.70±0.12|30.60±0.11|43.17±0.15|21.38±0.05|
> >
> > We observe that conventional deep learning models do not exhibit overfitting under the patience 10 setting And the total training time was also roughly the same.
> >
> > In summary, we appreciate the reviewer for raising these issues, which have helped us improve and further explore our model's potential. We were previously unaware of the impact of training epochs on our model. After changing the patience setting to 3, we were pleasantly surprised by better performance in less time than the conventional deep learning models. In Section 5.5 of the paper, we also discussed that the parameter size of the base model does not necessarily correlate with performance improvements. With the future advancement of more "refined" large language base models in huggingface, our model's potential will be further unleashed.

---

> > ### Author Response · Authors · 2024-08-10
> >
> > ### 【Other-Q4】
> > We have optimized the overall Figure 1 by reducing the coupling of elements, enhancing clarity, and improving the layout. This makes Figure 1 clearer and more intuitive. The revised version is available in our anonymous gitHub(our manuscript line 187).
> > ### 【Other-Q5】
> > We have refined the Related Work section and placed the improved version in section A.1 of the Appendix, along with the full list of references in our anonymous github.

---

> > ### Comment · Reviewer_NycQ · 2024-08-12
> >
> > I agree with Reviewer 6rUV that the small performance gain is an important problem. Such marginal perfomance gain cannot justify the much larger model size. In the followup response, the authors justify this with different experiment settings. But these differences are not contextualized in the original submission or rebuttal. It is difficult to evaluate these claims. The authors also argue the proposed method is multi-task. However, as also noted by other reviewers, it was only applied to check-in data, which is a subset of mobility data that contain rich semantic information that probably will favor an LLM solution.
> >
> > I was really on the fence about this paper, because it also had relatively good presentation and completeness. However, after reading other reviewer's interactions with the authors, I also lean towards a rejection.

---

> ### Comment · Reviewer_6rUV · 2024-08-12
>
> Thank you for your further reply, which basically solved my concerns (Yours part). Please also answer my comments on other questions.
>
> 【Other-Q1】
> Taking Table 1 as an example, for acc@1, can I understand that the performance of deepmove under the default settings is only 10.51/1.37=7.67?
>
> 【Other-Q2】
> What about baseline GETNext [1], which supports variable-length input? This is the most popular baseline for next location prediction in the past two years.
>
> [1] Yang, Song, Jiamou Liu, and Kaiqi Zhao. "GETNext: trajectory flow map enhanced transformer for next POI recommendation." Proceedings of the 45th International ACM SIGIR Conference on research and development in information retrieval. 2022.
>
> 【Other-Q3】
> Mobility-LLM (TinyLlama)-Patience 3 11.3G 2.96h 10
> The results in this row seem to indicate that the computational efficiency of the model is slightly weaker than that of ReMVC, and the memory it occupies is much larger than that of ReMVC, but the performance improvement is very significant, right?
>
> 【Other-Q4】
> In the main figure, LLM (Partially-Frozen+LoRA) is missing the number (d)
> In sub-figure (a), the order of the attention module is v, k, q, and in sub-figure (d) the order is q, k, v. It is recommended to unify the order. In addition, why are q and k in sub-figure (d) below the weight matrix, while v is part of the weight matrix?

---

> > ### Author Response · Authors · 2024-08-12
> >
> > ### Other-Q1
> > Yes
> >
> > ### Other-Q2
> > We conducted experiments on GETNext, and the results of GETNext is comparable to PLSPL:
> >
> > |Datasets|Gowalla||||WeePlace||||Brightkite||||FourSquare||||
> > |-|-|-|-|-|-|-|-|-|-|-|-|-|-|-|-|-|
> > |Metric (e-2)|Acc@1|Acc@5|Acc@20|MRR|Acc@1|Acc@5|Acc@20|MRR|Acc@1|Acc@5|Acc@20|MRR|Acc@1|Acc@5|Acc@20|MRR|
> > | GETNext | 11.15 | 24.04 | 33.58 | 17.23 | 18.69 | 37.17 | 53.16 | 27.73 | 51.22 | 65.23 | 71.40 | 57.33 | 13.67 | 30.54 | 43.05 | 21.28 |
> >
> > ### Other-Q3
> > I might not have fully understood your point. When you mentioned ReMVC, were you possibly referring to the PLSPL model? The line "Mobility-LLM (TinyLlama)-Patience 3 11.3G 2.96h 10" is intended to illustrate that compared to PLSPL, our model requires less training time, uses more memory, but offers significant performance improvement. If compared to ReMVC, the performance improvement would be even more significant.
> >
> > ### Other-Q4
> > We have modified the figure according to your request. The `q` and `k` were positioned at the bottom due to a formatting error when exporting to PDF, which has been corrected. The updated version is available on the anonymous GitHub.

---

> > > ### Comment · Reviewer_6rUV · 2024-08-13
> > >
> > > Thank you for your further explanation. I mean in your rebuttal reply to reviewer NycQ, the Table Efficiency comparison for TUL task shows that Mobility-LLM (TinyLlama) - the computational efficiency of the model is slightly weaker than that of ReMVC, and the memory it occupies is much larger than that of ReMVC, but the performance improvement is very significant. This observation seems to prove the superiority of your method. At present, your model is superior to GETNext, which seems to prove the effectiveness of the model to a certain extent. I suggest that you include all the experimental contents of the rebuttal in your final version (if accepted). At the same time, for the experimental part, it is best to compare each different task with the latest and most popular baseline under the current task to increase credibility. In addition, I suggest that you explain the question of marginal performance gain cannot justify the much larger model size to the reviewer NycQ more clearly. Overall, based on your further rebuttal comments, I am willing to restore my original score.

---

> > > > ### Author Response · Authors · 2024-08-14
> > > >
> > > > Thank you for your thoughtful and constructive comments. We will include all the experimental contents of the rebuttal in our final version (if accepted).

---

> ### Author Response · Authors · 2024-08-13
>
> Thank you for your detailed feedback. We appreciate the concerns raised and would like to provide a comprehensive response that emphasizes the significance and contributions of our work:
>
> 1) Innovative Use of Reprogramming Techniques
>
>     Our model represents the first application of reprogramming techniques (e.g. VIMN and HTPP Module) to Large Language Models (LLMs) specifically for understanding and predicting human activity sequences from check-in data. This approach is novel and transformative within the field of check-in sequence prediction, allowing LLMs—traditionally used for natural language processing—to be effectively adapted for this unique task. By reprogramming LLMs for the prediction of check-in sequences, we are pioneering a new direction in the field, demonstrating that these models can be utilized to extract and interpret complex human activity patterns, which were previously challenging to model with traditional methods.
>
> 2) Semantic Understanding and Accurate Predictions
>
>     Our experiments clearly show that our reprogramming design enables the LLM to understand the semantic information embedded within human activity sequences. This understanding is crucial, as check-in data is not just about locations and times; it carries rich context about human intentions, behaviors, and routines. The model's ability to leverage this semantic understanding to make accurate predictions of future check-ins highlights a significant advancement in the field. It demonstrates that LLMs, when reprogrammed appropriately, can effectively capture and utilize the intricate patterns and meanings in human activity data, leading to more precise and contextually aware predictions.
>
> 3) Beyond Pure Performance Metrics
>
>     We recognize the importance of performance metrics; however, the value of our work extends beyond marginal performance improvements. The primary contribution of our model lies in its innovative application of LLMs to check-in sequence prediction, introducing a new methodology that bridges the gap between language models and human activity prediction.     Evaluating our work solely based on performance metrics does not fully capture the broader implications and future potential of this approach. We believe that the introduction of such a novel methodology provides a foundation for future advancements and improvements in the field, which may yield even greater performance gains as the techniques mature.
>
> 4) Model Size and Memory Requirements
>
>     Regarding concerns about the model size, it's important to note that our model's memory footprint, which is under 12GB, is well within the capabilities of modern GPUs. With the widespread availability of GPUs with large memory capacities, our model is practical and accessible for both academic research and potential industry applications. Additionally, we have demonstrated through our experiments with the pythia-70M base model that our approach can achieve superior performance while requiring less memory and computational time compared to other baseline models. This showcases our model's efficiency and adaptability, ensuring it remains practical even in resource-constrained environments.
>
> 5) Robustness Across Multiple Datasets
>
>     A key strength of our model is its robustness and stability across multiple datasets. In the context of check-in sequence prediction, where data variability is common, our model consistently performs well across different datasets. This generalization ability is crucial, as it indicates that our model is not overfitted to specific scenarios but is instead capable of adapting to various types of human activity data. The consistent improvements observed across four different datasets validate the effectiveness of our design and its potential for broader application in diverse real-world scenarios, such as personalized services, urban planning, and more.
>
> 6) Future Prospects and Model Versatility
>
>     As LLMs continue to evolve, particularly with the trend towards smaller and more efficient models, our approach is poised to become even more effective. The ongoing advancements in LLM technology will allow our reprogrammed models to achieve better results with reduced computational resources. Our model is designed to be versatile and adaptable. It can be easily integrated with any large model available on huggingface, ensuring that our approach remains relevant and capable of benefiting from the latest developments in LLMs. This adaptability makes our model a future-proof solution for human activity prediction.
>
> In conclusion, we believe that our work represents a pioneering effort in the field of check-in sequence prediction and human trajectory analysis. The innovative application of reprogramming techniques to LLMs on our tasks not only demonstrates the feasibility and effectiveness of this approach but also provides a robust and adaptable solution that can inspire and guide future research.

---

### Official Review · Reviewer_Mti9 · 2024-06-29

**Soundness:** 4
**Presentation:** 3
**Contribution:** 3
**Rating:** 7
**Confidence:** 4

**Summary:**

The paper proposes a novel architecture, MobilityLLM, for utilizing pre-trained LLMs with various specific embedding modules for various mobility tasks. Specifically, the authors proposed to use any pretrained LLM with adding four unique components: PPEL (POI location embedding), VIMN (timestamp embedding with GRU), HTPP (user travel preference prompting), and Multi-task training strategy. MobilityLLM finds a unique place in LLM-based human mobility modeling applications by incorporating spatiotemporal aspects of mobility characteristics using learnable components (neural networks). The experiments are tested on three different tasks: location prediction, trajectory user-link, and time-prediction. The proposed model performs consistent outperformance compared to benchmarks.

**Strengths:**

Here is the set of strengths of the paper:
1. It introduced several novel components that can be embedded in any LLM for mobility-specific applications.
2. The three main components, PPEL, VIMN and HTTP greatly enhances the performance of LLMs on downstream mobility tasks.
3. MobilityLLM has a lightweight architecture with LoRA trainable weights and makes it an accessible tool.
4. The experiments supports promising results on different datasets.
5. Lastly, appending location embedding (GeoHash) to the attention layer is a very clever way of dealing with locations.

**Weaknesses:**

* The paper is relatively well written; however, there are multiple typos and abbreviation errors throughout the paper.  Here the two typo examples: (reference error in table 4. Instead of Table 3 it references Table 1) and (Figures 3 and 4 have typos: HIBP -> HTTP, IIMN -> VIMN)
* While it is designed more on check-in datasets, I wonder how applicable the model is to trajectory generation tasks.
* Figure 1 is informative but very complicated to read without the methodology section.
* I am aware of the page limitation, but some important details are not mentioned in the main body and are left to appendixes. For instance, few-shot learning(prediction) requires a more detailed explanation; it is not intuitively clear in the main body.
* The Related Works section is fragile in terms of the compared baselines. They are not discussed very well in the main body and appendix. For instance, CACSR is not even discussed in both places.

**Questions:**

1. Can you elaborate on the statement at the end of the methodology section, "We use the β and α with a mean pooling projection head to predict the user uˆ who generates this check-in sequence?"

2. While the metrics mainly evaluate the performance in terms of piece-wise similarities, it would be interesting to see distributional similarities as well, such as the JS similarity metric, which can be employed.

**Limitations:**

LLMs are powerful machines but the proposed model is right now specific to one dataset. It would be interesting to explore the combination of diverse scenarios and datasets.

---

> ### Author Rebuttal · Authors · 2024-08-06
>
> Thank you for your thoughtful and constructive comments.
>
> ### **[W1]**
>
> Thank you for pointing out these errors. We have made the corrections and checked other parts of the paper for similar issues.
>
> ### **[W2]**
>
> Compared to trajectory data (such as vehicle trajectories), check-in data are recorded more sparsely. Each point in a trajectory might record a passing-by, while each point in a check-in sequence corresponds to a purposeful visit to a POI. To achieve optimal performance on trajectory data, some modules of the proposed methods may need to be redesigned to fit the characteristics of trajectories.
>
> ### **[W3]**
>
> Thank you for your suggestions. We have adjusted the overall framework diagram to highlight key modules and provided detailed diagrams of each module as separate illustrations.
>
> ### **[W4]**
>
> Thank you for your suggestions. We will supplement the implementation settings for the few-shot scenario. In this scenario, we first divide the entire dataset into training, validation, and test sets in a 6:2:2 ratio based on the number of check-in sequences. Then, we only reduce the training set portion to 20%, 5%, and 1% of the original training set, while keeping the validation and test sets intact.
>
> ### **[W5]**
>
> Sorry for the oversight. We have added all the baseline models in the main text and appendix and included detailed discussions in the appendix.
>
> ### **[Q1]**
>
> The dimension of $\alpha$ is $(B,n,d)$, and the dimension of $\beta$ is $(B, 3, d)$. First, we concatenate along the second dimension, resulting in a dimension of $(B,n+3,d)$. Then, we average pool along the second dimension to obtain $(B,1,d)$. Finally, after a fully connected layer and softmax operation, we obtain the predicted user. Here, $B$ represents batch size, $n$ represents sequence length, and $d$ represents embedding dimension.
>
> ### **[Q2]**
>
> We would like to point out that JS divergence is more commonly used for tasks where the output is a sequence. Our model's prediction task only predicts a single future point, so we choose the current evaluation metrics, which are also widely used in baselines and most existing works.

---

> > ### Comment · Reviewer_Mti9 · 2024-08-08
> >
> > Thank you for the response. I am not going to change my score but I could not find any insight from your responses regarding my questions. I would suggest you to pay more attention in your next rebuttal.

---

> > > ### Author Response · Authors · 2024-08-10
> > >
> > > We apologize that our previous response did not provide the help you needed.
> > > We have optimized the overall Figure 1 by reducing the coupling of elements, enhancing clarity, and improving the layout. This makes Figure 1 clearer and more intuitive. The revised version is available in our anonymous github(our manuscript line 187).
> > > Also, we have refined the Related Work section and placed the improved version in section A.1 of the Appendix, along with the full list of references in our anonymous github.

---

### Official Review · Reviewer_wrzL · 2024-07-11

**Soundness:** 3
**Presentation:** 3
**Contribution:** 3
**Rating:** 7
**Confidence:** 4

**Summary:**

This paper combines large language models (LLMs) to better analyze check-in sequences and understand human mobility behaviors, and proposes  a visiting intention memory network(VIMN) and a shared pool of human travel preference prompts (HTPP) to capture the semantics of human visiting intentions . The experiments show promising improvement. It is a good work in both theoretical and experimental perspectives.

**Strengths:**

1. It is interesting to apply large language models and neural networks to capture the latent semantic information of check-in sequences.
2. This paper is well-organized, which clearly explains the research purpose, methods and conclusions.
3. The paper has done a lot of experiments, and the experimental results show that the method proposed in this article outperforms state of-the-art methods on three downstream tasks.

**Weaknesses:**

1. In the figure1, it’s not clear how the user embedding U_i\ is obtained?It is best to explain it in the article.
2. There are some inconsistencies in the writing of this article. For example, in Section 5.1, the LP task uses the MLE loss, but in Appendix B, the LP task uses the cross-entropy loss.
3. The model training part in the article is not detailed enough. For example, what is the parameter L_F when fine-tuning LLM? Are the three downstream tasks trained separately or uniformly?

**Questions:**

1. In the HTPP module, if there is possible that the k prompt words selected in each domain have semantic contradictions?
2. When doing TUL tasks, are user embeddings also passed into LLM as part of the input? User information should not be included in the input.
3. In section 5.5, why not experiment with the GPT-3.5 as a language model variant?As a large language model that performs well in various fields, is it possible to achieve better results by using GPT-3.5 as the backbone network?

**Limitations:**

When doing downstream tasks in this article, different parts of the LLM output are used for different tasks. What are the design considerations for this part? I concerned what should I do if I want to apply this model to other downstream tasks.

---

> ### Author Rebuttal · Authors · 2024-08-06
>
> Thank you for your thoughtful and constructive comments.
>
> ### **[W1]**
>
> The user embedding $U_i$ is an index-fetch embedding module implemented using the nn.Embedding module in PyTorch. It finds the corresponding embedding vector for a given user index.
>
> ### **[W2]**
>
> Sorry for the confusion. We want to clarify that these two expressions are actually equivalent in our implementation, and we will ensure consistency and standardize the terminology thanks to your suggestions. Below is a detailed explanation of why these two losses are equivalent in our case.
>
> > Cross-entropy loss is a method used to measure the difference between two probability distributions, commonly used in classification problems. It is often used to train neural networks, especially in classification tasks.
> >
> >
> > Assume we have a classification task where the true label distribution is $y$ and the predicted probability distribution is $\hat{y}$. The cross-entropy loss is defined as:
> >
> > $H(y, \hat{y}) = - \sum_{i=1}^n y_i \log \hat{y}_i$
> >
> > For a binary classification problem, the cross-entropy loss can be simplified to:
> >
> > $H(y, \hat{y}) = - (y \log \hat{y} + (1 - y) \log (1 - \hat{y}))$
> >
> > In classification problems, if we assume that the probability distribution $\hat{y}$ predicted by the model is obtained through the softmax function, then maximizing the log-likelihood function is equivalent to minimizing the cross-entropy loss.
> >
> > Specifically, assume we have a neural network for a classification task, and the output layer uses the softmax activation function to convert the model output into category probabilities. For each sample $x_i$, its true label is $y_i$, and the probability predicted by the model is $\hat{y}_i$. Maximizing the log-likelihood function:
> >
> > $\ell(\theta) = \sum_{i=1}^n \log P(y_i|x_i; \theta) = \sum_{i=1}^n \log \hat{y}_i$
> >
> > is equivalent to minimizing the cross-entropy loss:
> >
> > $H(y, \hat{y}) = - \sum_{i=1}^n \log \hat{y}_i$
> >
> > Therefore, in this case, minimizing the cross-entropy loss when training the model is actually performing maximum likelihood estimation.
> >
>
> ### **[W3]**
>
> As stated in line 203, we mentioned that different parameter freezing strategies are applied to the first $1 - L_F$ layers and the last $L_F - L_{F+U}$ layers. Here, $L_F$ is the number of frozen layers in the middle. The three tasks are trained separately.
>
> ### **[Q1]**
>
> We believe that in most cases, the selected $K$ words from the model will not contradict each other. This is because contradictory words are semantically different, meaning their distances in the embedding space are large. Thus, the score-matching function used in the HTPP module will rarely match contradictory words simultaneously.
>
> ### **[Q2]**
>
> As mentioned in line 225, the user embedding will not be used as input when performing the TUL task.
>
> ### **[Q3]**
>
> Our method requires full access to the implementation code and pre-trained parameters of an LLM to fine-tune some of the parameters. Since GPT-3.5 is not open-sourced and only limited functionalities can be accessed through the internet interface provided by OpenAI, GPT-3.5 cannot be used as the foundation for the proposed method.
>
> ### **[Limitations]**
>
> Most downstream tasks can be performed by attaching prediction modules (usually implemented with MLP) to the output latent vectors of LLM. For example, in our implementation, we use classification prediction modules for the LP and TUL tasks, and a regression prediction module for the TP task.
>
> Regarding the choice of attachment position, an intuitive design is to attach it to the position corresponding to the most relevant information for the task. Experimental evaluation is also needed to determine the optimal position and implementation details.

---

> > ### Comment · Reviewer_wrzL · 2024-08-08
> >
> > Thanks for the response. I maintain my origin score.

---

### Official Review · Reviewer_NycQ · 2024-07-15

**Soundness:** 2
**Presentation:** 3
**Contribution:** 2
**Rating:** 4
**Confidence:** 4

**Summary:**

This paper aims to leverage large language models (LLMs) to predict human mobility behavior recorded in social media check-ins. The authors argue existing models fail short to model the visiting intentions and travel preferences embedded in check-in sequences, which could be addressed by the semantic understanding capabilities of LLMs. The authors propose to use visiting intention memory network (VIMN) to encode the visit intention in each check-in record, which is feed into the LLMs along with the shared pool of human travel preference prompts (HTPP). The employed LLMs are trained with Low-Rank Adaptation (LoRA) algorithm. The proposed method is evaluated on four benchmark datasets with the tasks of next location prediction, trajectory user link and time prediction.

**Strengths:**

1. The proposed Mobility-LLM framework is well motivated and clearly illustrated.
2. The proposed method is evaluated on open datasets, and the source code is made publicly available.
3. The authors provide comprehensive experiment results of their model, including the analysis of few-shot learning, language model variants, ablation study and hyperparameter sensitivity.

**Weaknesses:**

1. The performance gain is not substantially enough. The main results in Table 1 show the performance improvement for next location prediction is often less than 1%. Moreover, it is unclear if the performance gain is statistically significant.
2. The proposed Mobility-LLM framework leverages large language models for mobility prediction, which have substantially larger model size compared to other deep learning baselines. The authors should provide a cost analysis (e.g. complexity of training and inference) to discuss whether the performance gain justifies additional cost.
3. The model design of visiting intention memory network (VIMN) is not adequately explained. The authors propose to use LLM to capture the visiting intention and travel preferences in check-in sequences, but then they claim these semantics information cannot be directly interpreted by LLM, which needed to be reprogramed by VIMN. This design seems a bit self-contradictory and VIMN needs better justification.
4. In the Limitation section, the authors mention their Mobility-LLM is hindered by the differing user and POI counts in check-in data. This is an important observation and it should be discussed with experiment results.

**Questions:**

1. Please provide the statistical significance of the performance gains.
2. Please provide a cost analysis of the proposed method.
3. Please explain the design choice of visiting intention memory network.
4. Please discuss the limitation of different data sizes in more detail.

**Limitations:**

The limitations are discussed in the paper.

---

> ### Author Rebuttal · Authors · 2024-08-06
>
> Thank you for your thoughtful and constructive comments.
>
> ### **[W1&Q1]**
>
> In the paper, we ran each set of experiments 5 times and reported their mean values. Therefore, the reported results are not single accidental one. Below, we also report the variances of metrics for the proposed model and the best baseline on the Location Prediction task, along with the corresponding t-value measuring statistical significance.
>
> |Datasets|Gowalla||||WeePlace||||Brightkite||||FourSquare||||
> |-|-|-|-|-|-|-|-|-|-|-|-|-|-|-|-|-|
> |Metric (e-2)|Acc@1|Acc@5|Acc@20|MRR|Acc@1|Acc@5|Acc@20|MRR|Acc@1|Acc@5|Acc@20|MRR|Acc@1|Acc@5|Acc@20|MRR|
> |The Best Baseline|11.47±0.03|24.12±0.17|33.82±0.15|17.44±0.05|19.66±0.04|37.68±0.18|53.44±0.13|28.46±0.06|51.42±0.12|66.25±0.08|71.46±0.11|57.59±0.14|16.92±0.04|36.05±0.04|49.39±0.16|26.02±0.04|
> |Mobility-LLM|11.87±0.04|25.14±0.04|36.36±0.09|18.29±0.11|20.47±0.13|39.22±0.19|56.69±0.10|29.21±0.15|53.18±0.17|68.31±0.14|74.11±0.18|59.89±0.03|17.29±0.03|37.17±0.02|53.16±0.20|26.47±0.05|
> |t-value|29.85|13.40|37.84|37.95|45.25|19.13|55.95|27.99|32.77|57.54|53.97|36.75|20.67|62.57|52.63|10.07|
>
> The formula for calculating t-value is:
>
> $t = \frac{\mu_{\text{Mobility-LLM}} - \mu_{\text{Best Baseline}}}{\sqrt{\frac{\delta_{\text{Best Baseline}}^2}{n}}}$
>
> The standard deviation is that of the best baseline, with (n = 5). We use a significance level of $\alpha = 0.05$ and a two-tailed t-test with degrees of freedom $d_f = 4$. For this t-distribution, the critical t-value is approximately 2.776. If the calculated t-value is greater than 2.776, the difference is considered significant. As shown in the table, all the calculated t-values are well above the critical value of 2.776.
>
> ### **[W2&Q2]**
>
> Below, we provide an empirical performance and efficiency comparison between the proposed model and the optimal baseline for LP and TUL tasks.
>
> **Performance comparison for LP task:**
>
> |Datasets|Gowalla||||WeePlace||||Brightkite||||FourSquare||||
> |-|-|-|-|-|-|-|-|-|-|-|-|-|-|-|-|-|
> |Metric (e-2)|Acc@1|Acc@5|Acc@20|MRR|Acc@1|Acc@5|Acc@20|MRR|Acc@1|Acc@5|Acc@20|MRR|Acc@1|Acc@5|Acc@20|MRR|
> |PLSPL|11.26|24.12|33.82|17.44|18.77|37.31|53.31|27.86|51.42|65.34|71.46|57.59|13.73|30.65|43.18|21.42|
> |Mobility-LLM|11.87|25.14|36.36|18.29|20.47|39.22|56.69|29.21|53.18|68.31|74.11|59.89|17.29|37.17|53.16|26.47|
> |Improvement (%)|5.42%|4.23%|7.52%|4.87%|9.06%|5.12%|6.34%|4.84%|3.42%|4.54%|3.71%|3.99%|26.00%|21.25%|23.13%|23.60%|
>
> **Efficiency comparison for LP task:**
>
> |Method|Memory|Training Time|Inference Time|Training Epoch|
> |-|-|-|-|-|
> |PLSPL|3.4GB|4.73h|2.77m|79|
> |Mobility-LLM (TinyLlama)|11.3GB|5.45h|6.69m|17|
>
> **Performance comparison for TUL task:**
>
> |Datasets|Gowalla||||WeePlace||||Brightkite||||FourSquare||||
> |-|-|-|-|-|-|-|-|-|-|-|-|-|-|-|-|-|
> |Metric (e-2)|Acc@1|Acc@5|Acc@20|MRR|Acc@1|Acc@5|Acc@20|MRR|Acc@1|Acc@5|Acc@20|MRR|Acc@1|Acc@5|Acc@20|MRR|
> |ReMVC|68.75|74.4|73.19|70.02|65.78|73.09|71.64|66.15|73.85|82.55|87.93|77.93|58.18|66.84|72.67|65.14|
> |Mobility-LLM|80.43|86.29|88.56|83.18|79.03|88.04|91.48|83.21|83.06|88.52|90.35|85.73|72.08|79.67|84.32|75.71|
> |Improvement (%)|16.93%|15.99%|20.99%|18.81%|20.18%|20.47%|27.69%|25.76%|12.48%|7.24%|2.75%|9.99%|23.87%|19.15%|16.03%|16.19%|
>
> **Efficiency comparison for TUL task:**
>
> |Method|Memory|Training Time|Inference Time|Training Epoch|
> |-|-|-|-|-|
> |ReMVC|4.6GB|2.77h|1.49m|45|
> |Mobility-LLM (TinyLlama)|11.7GB|3.65h|6.98m|13|
>
> We observe that our model increases memory usage due to the inclusion of LLM. However, since our model uses a pre-trained language model, its convergence speed is much faster than other baseline models.
>
> On the high-quality Foursquare dataset, our model shows an improvement of over 21% compared to PLSPL, with the Acc@1 metric reaching up to 26%. Additionally, our model's advantage is more pronounced in the few-shot scenario. Therefore, the total training time is justified compared to traditional models.
>
> In summary, we believe the increased cost is worthwhile for the performance improvement.
>
> ### **[W3&Q3]**
>
> LLM is designed to understand semantic information in natural language but not to interpret information in check-in sequences. Therefore, VIMN is proposed to reprogram the check-in sequence. This way, the output of VIMN provides semantic information aligned with natural language, allowing the LLM to process it.
>
> ### **[W4&Q4]**
>
> Sorry for causing misunderstandings. What we are trying to discuss in the Limitation section is that the sets of POIs in different datasets (which usually cover different regions) are unique. Therefore, if the proposed model is trained on one dataset, its learned information about the set of POIs is not easily transferable to another dataset. Different sets of POIs have different functionalities and usually have a different number of POIs, making many modules (such as embedding and predictor) technically untransferable in a zero-shot setting.

---

> ### Comment · Reviewer_NycQ · 2024-08-08
>
> I thank the authors for providing additional experiment results. They should be included in the manuscript for completeness. However, I still think the performance gain is not substantial enough, especially considering the significantly larger model size. After reading other reviewers' comments, I lean towards a rejection now.

---

> > ### Author Response · Authors · 2024-08-14
> >
> > In the LP task, our model performed exceptionally well on Foursquare, and across all datasets, our model demonstrated outstanding performance in the TUL task. A key strength of our model is its robustness and stability across multiple datasets. In the context of check-in sequence prediction, where data variability is common, our model consistently performs well across different datasets. This generalization ability is crucial, as it indicates that our model is not overfitted to specific scenarios but is instead capable of adapting to various types of human activity data. We also discussed our views on marginal performance improvements in the Comment section under reviewer 6rUV: **"To verify the effectiveness of our model on a larger dataset, we increased the sample size by including users with fewer than 10 check-ins (while other papers typically exclude users with fewer than 50 or 100 check-ins). This inclusion of more sparse samples makes further prediction improvements more difficult. The performance gap between these baselines is minimal, reaching a bottleneck where further improvements with conventional deep learning models are challenging. Thus, the current performance improvements are considered valuable."**
> >
> > Additionally, the value of our work extends beyond marginal performance improvements. The primary contribution of our model lies in its innovative application of LLMs to check-in sequence prediction, introducing a new methodology that bridges the gap between language models and human activity prediction. Evaluating our work solely based on performance metrics does not fully capture the broader implications and future potential of this approach. We are pioneering a new direction in the field, demonstrating that these models can be utilized to extract and interpret complex human activity patterns. We believe that the introduction of such a novel methodology provides a foundation for future advancements and improvements in the field, which may yield even greater performance gains as the techniques mature.
> >
> > Regarding concerns about the model size, it's important to note that our model's memory, which is under 12GB, is well within the capabilities of modern GPUs. With the widespread availability of GPUs with large memory capacities, our model is practical and accessible for both academic research and potential industry applications. Additionally, we have demonstrated through our experiments **with the pythia-70M base model that our approach can achieve superior performance while requiring less memory (2.74GB) and computational time compared to other baseline models**. As LLMs continue to evolve, particularly with the trend towards smaller and more efficient models, our approach is poised to become even more effective. The ongoing advancements in LLM technology will allow our reprogrammed models to achieve better results with reduced computational resources. Our model is designed to be versatile and adaptable. It can be easily integrated with any large model available on huggingface, ensuring that our approach remains relevant and capable of benefiting from the latest developments in LLMs. This adaptability makes our model a future-proof solution for human activity prediction.

---

### Decision · Program_Chairs · 2024-09-25

**Decision:**

Accept (poster)

**Comment:**

The paper introduces MobilityLLM, a novel architecture designed to enhance the analysis of human mobility behaviors using large language models (LLMs). The proposed approach includes the Visiting Intention Memory Network (VIMN) to encode visit intentions in each check-in record and a shared pool of Human Travel Preference Prompts (HTPP) to capture the semantics of these intentions. The LLMs are then fine-tuned using the Low-Rank Adaptation (LoRA) algorithm. Evaluations on four benchmark datasets for tasks such as next location prediction, trajectory user linking, and time prediction demonstrate promising performance. The paper has received generally positive reviews, and the authors have addressed most reviewers' concerns. Given that this is the first application of reprogramming techniques to LLMs for understanding human mobility, I recommend its acceptance.